# Diffusion Network Inference for Cross-layer Cascades

**Siyu Huang**\*
Department of Statistics
The Pennsylvania State University
`sph6006@psu.edu`

**Yubai Yuan**\*†
Department of Statistics
The Pennsylvania State University
`yvy5509@psu.edu`

**Abdul Basit Adeel**
Department of Sociology and Criminology
The Pennsylvania State University
`axa6372@psu.edu`

## Abstract

A cascade over a network refers to the diffusion process where behavior changes occurring in one part of an interconnected population lead to a series of sequential changes throughout the entire population. In recent years, there has been a surge in interest and efforts to understand and model cascade mechanisms since they motivate many significant research topics across different disciplines. The propagation structure of cascades is governed by underlying diffusion networks that are often hidden. Inferring diffusion networks thus enables interventions in cascading process to maximize information propagation and provides insights into the Granger causality of interaction mechanisms among individuals. In this project, we propose a novel double network mixture model for inferring latent diffusion network in presence of strong cascade heterogeneity. The new model represents cascade pathways as a distributional mixture over diffusion networks that capture different cascading patterns at the population level. We develop a data-driven optimization method to infer diffusion networks using only visible temporal cascade records, avoiding the need to model complex and heterogeneous individual states. Both statistical and computational guarantees are established for the proposed method. We apply the proposed model to analyze research topic cascades in social sciences across U.S. universities and uncover the latent research topic diffusion network among top U.S. social science programs.

## 1 Introduction

Cascades over network refer to the diffusion processes where behavior changes in a part of an interconnected population lead to a series of sequential changes throughout the entire population. In recent years, there are surging interests and efforts to understand and model the cascade mechanism since it motivates many significant research topics in different areas, including social influence [12, 13, 20], information propagation via social media [32, 1], diffusion of policy and social norms [4, 33], viral marketing [25, 9], and contagion of infectious diseases [21, 35].

One fundamental problem is to understand the diffusion networks that govern cascade propagation patterns. However, diffusion networks are often hidden and need to be inferred from observed cascading behaviors. For example, in the case of infectious diseases, we can observe when an individual is infected but need to impute the missing information on who infects this individual.

---

\*Co-first authors.
†Corresponding author.

39th Conference on Neural Information Processing Systems (NeurIPS 2025).

More importantly, real-world cascading behaviors often exhibit strong heterogeneity and are jointly governed by different diffusion patterns. For example, in information cascade and epidemiology, cascades can diffuse among population via different transmission channels building on various social relations, and thus lead to heterogeneous propagation speeds and scales [24, 30]. Furthermore, cascading heterogeneity originates from the variability of individuals' statuses in engaging the cascade [39, 11]. In social media like X and Instagram, the spreading pattern and the speed of messages heavily depend on users' activity status. An active user will respond more instantly to interesting messages and accelerate the information cascade compared to inactive users [10]. As for volatility cascades in financial markets, structure heterogeneity in volatility diffusion depends on different time horizons of the agents in the market [41]. In these scenarios, individual statuses determine the transmission channels engaging in the cascades, and changes of individual statuses can also change the downstream cascade diffusion patterns. To conclude from these examples, diffusion patterns can exhibit combinatorial complexity as population grows. We present an example of cascade diffusion when transmission channels of individuals vary in Figure (1).

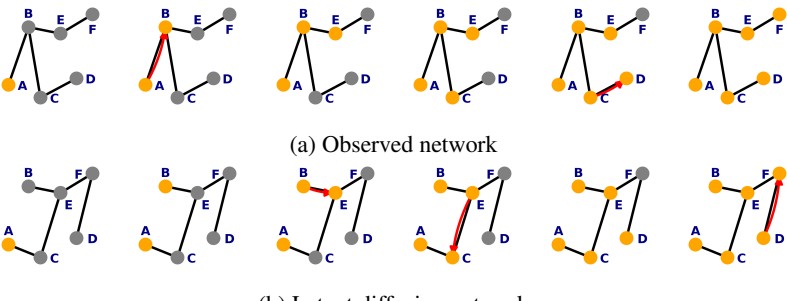

(a) Observed network

(b) Latent diffusion network

Figure 1: A cascade diffuses through nodes $\{A, B, C, D, E, F\}$ on a two-layer network. Yellow nodes: activated nodes; gray nodes: inactivated nodes. $\{B, D\}$ are activated via the observed network and $\{E, C, F\}$ are activated via the latent network.

In this paper, we propose a novel double network mixture model to infer multiple diffusion networks simultaneously from heterogeneous cascade data. The proposed model introduces a distributional mixture of diffusion networks to capture the heterogeneous cascading patterns, where diffusion networks provide complementary connection information. The main advantage of the distributional mixture is to avoid modeling the complex individual status changes. Specifically, the proposed model can describe the diffusion process over multi-layer networks where cascades propagate across different layers alternatively. Compared to existing methods, the proposed method can uncover latent diffusion networks even when the number of diffusion patterns is exponential to the number of nodes. Furthermore, the parameter estimation in our model can be solved by a sequence of convex optimization problems, which leads to both statistical and computational guarantees for our diffusion network estimation.

## 2   Related works

Various directed probabilistic graphical models have been developed to infer diffusion networks from observed cascade samples [17, 19, 31, 8, 22, 18]. Generally, these models treat infection time as a continuous random variable and construct the likelihood of cascade samples based on the target diffusion network under local Markov assumption. To capture heterogeneous diffusion patterns, several multi-pattern cascade models have been proposed [37, 40], where cascade samples are adaptively clustered into groups and each group corresponds to a distinct diffusion network. Among these methods, ConNIe [29], NetRate [31], and MMRate [37] are popular representatives. Specifically, ConNIe employs a maximum likelihood formulation via convex programming, incorporating an $l_1$-type penalty to promote sparsity in the inferred network. Building upon ConNIe, NetRate explicitly represents diffusion as a continuous-time probabilistic process, characterized by edge-specific transmission rates governing edge-wise diffusion probabilities. MMRate further extends this framework by accommodating multiple distinct diffusion patterns, assuming multiple heterogeneous latent networks, with each cascade diffusing via one network according to a certain probability.

# 3 Methodology

## 3.1 Continuous-time cascade on single network

Consider a network with $N$ nodes where each node has two infection conditions in a single cascade: infected (activated) and uninfected (inactivated), and an infected node will always remain infected. A cascade is a $N$-dimensional temporal record $\boldsymbol{t} = (t_1, t_2, \cdots, t_N)$, where $t_i$ is the infection time of the $i$-th node. Instead of infinite time horizon, we observe a cascade within a finite time window of length $T$, i.e., $\forall i, t_i \in [t_0, t_0 + T]$, where $t_0 := \min_{1 \leqslant i \leqslant N} \{t_i\}$ is the infection time of source node. We denote the infection time of the nodes not infected in the observation window as $t_0 + T$. Without loss of generality, we assume $t_0 = 0$ in this paper.

A cascade diffuses nodewisely over edges of a diffusion network. The continuous-time model formulates the cascade transmission from an infected node $j$ to another infected node $i$ with survival analysis models. Specifically, given a node $j$ being infected at time $t_j$, let $f(t_i \mid t_j, \lambda_{ji})$ denote the likelihood of node $i$ being infected by node $j$ at time $t_i$, where $t_i \geqslant t_j$. The transmission rate $\lambda_{ji}$ represents how fast cascades diffuse from node $j$ to node $i$. Accordingly, the cumulative probability function is $F(t_i \mid t_j, \lambda_{ji}) = \int_{t_j}^{t_i} f(s \mid t_j, \lambda_{ji}) ds$. We consider the hazard rate function defined as

$$H(t_i \mid t_j, \lambda_{ji}) = \frac{f(t_i \mid t_j, \lambda_{ji})}{1 - F(t_i \mid t_j, \lambda_{ji})} = \frac{f(t_i \mid t_j, \lambda_{ji})}{S(t_i \mid t_j, \lambda_{ji})}, \ S(t_i \mid t_j, \lambda_{ji}) = \exp\Big( - \int_{t_j}^{t_i} H(t \mid t_j) dt \Big),$$

where $S(t_i \mid t_j, \lambda_{ji}) = 1 - F(t_i \mid t_j, \lambda_{ji})$ is the survival function that indicates the probability of node $i$ not being infected by node $j$ until time $t_i$. Typical parametric forms of hazard rate functions are Exp model: $H(t_i \mid t_j, \lambda_{ji}) = \lambda_{ji}$; Pow model: $H(t_i \mid t_j, \lambda_{ji}) = \lambda_{ji} \frac{1}{t_i - t_j}$; Ray model: $H(t_i \mid t_j, \lambda_{ji}) = \lambda_{ji}(t_i - t_j)$. Given that $\lambda_{ji}$ controls the likelihood and speed of transmission between node $j$ to node $i$, the global cascading pattern over network can be captured by matrix $\boldsymbol{\Lambda} = (\lambda_{ij}) \in \mathrm{R}_+^{N \times N}$, where rows represent sender nodes and columns represent receiver nodes. Note that $\boldsymbol{\Lambda}$ can be asymmetric, i.e., $\lambda_{ij} \neq \lambda_{ji}$, when node $i$ and $j$ are different in the capacity of launching transmission.

The diffusion process is typically modeled by independent cascade models [22], where any infected nodes can infect a node independently and a node stays infected once another node infects it. Therefore, one can formulate the likelihood of node $i$ being infected by potential parent nodes $\{j : t_j < t_i\}$ at time $t_i$ as

$$\boldsymbol{P}_I(t_i; \boldsymbol{\Lambda}_{\cdot i}) := \boldsymbol{P}(t_i \mid \{t_j : t_j < t_i\}) = \sum_{j : t_j < t_i} f(t_i \mid t_j, \lambda_{ji}) \prod_{k : t_k < t_i, k \neq j} S(t_i \mid t_k, \lambda_{ki}) \quad (1)$$

$$= \sum_{j : t_j < t_i} H(t_i \mid t_j, \lambda_{ji}) \prod_{k : t_k < t_i} S(t_i \mid t_k, \lambda_{ki})$$

On the other hand, if node $i$ is not activated by any parent nodes in the observation window, then the corresponding likelihood for $\{i : t_i = T\}$ is

$$\boldsymbol{P}_U(T; \boldsymbol{\Lambda}_{\cdot i}) := \boldsymbol{P}(t_i \mid \{t_j : t_j < t_i\}) = \prod_{j : t_j < T} S(T \mid t_j, \lambda_{ji}), \quad (2)$$

Since infections of each node are conditionally independent given the corresponding parent nodes, we can decompose the likelihood of a cascade $\boldsymbol{t}$ as the multiplication of a series of conditional probabilities:

$$\boldsymbol{P}(\boldsymbol{t}; \boldsymbol{\Lambda}) = \prod_{i=1}^{N} \boldsymbol{P}(t_i \mid \{t_j : t_j < t_i\}; \boldsymbol{\Lambda}_{\cdot i}) = \prod_{i=1}^{N} \big\{ \mathbb{1}(t_i < T) \boldsymbol{P}_I(t_i; \boldsymbol{\Lambda}_{\cdot i}) + \mathbb{1}(t_i \geqslant T) \boldsymbol{P}_U(T; \boldsymbol{\Lambda}_{\cdot i}) \big\}$$

Therefore, the full likelihood of independent cascade samples $\{\boldsymbol{t}^{(c)}\}_{c=1}^{C}$ is $\prod_{c=1}^{C} \boldsymbol{P}(\boldsymbol{t}^{(c)}; \boldsymbol{\Theta})$. The advantage of the above model is that each cascade sample induces a directed acyclic graph, whose local Markov property allows the decomposition of likelihood and thus a reduction in computational complexity of inferring the diffusion network $\boldsymbol{\Theta}$.

## 3.2 Double network mixture model

In many applications such as social media, there exist multiple networks among an interconnected population that reflect different types of relations. Cascade diffusion pattern can change over different networks, and cascade can proceed on different networks alternatively and simultaneously due to the inter-layer interactions. The diffusion process on multi-layer networks can greatly increase the degree of freedom in possible diffusion pathways, which leads to strong heterogeneity in cascade observations. To model the multi-network cascade behavior, we propose the **double network mixture model**. Consider two diffusion networks $\boldsymbol{\Theta}$ and $\boldsymbol{\Psi}$ among the same population with $N$ units, where $\boldsymbol{\Theta}_{ij}$ and $\boldsymbol{\Psi}_{ij}$ are the transmission rate from node i to node j corresponding to two relations. We introduce double diffusion indicators $Z_i^c \in \{0, 1\}$ such that

$$Z_i^c \overset{\text{i.i.d}}{\sim} \text{Bern}(\pi_i), \ i = 1, \cdots, N, \ c = 1, \cdots, C.$$

For cascade $c$, node $i$ is activated via diffusion pathway on network $\boldsymbol{\Theta}$ if $Z_i^c = 1$, and $Z_i^c = 0$ if via network $\boldsymbol{\Psi}$. Denote $\boldsymbol{\pi} = \{\pi_i\}_{i=1}^N \in [0, 1]^N$ where $\pi_i$ is the probability of node $i$ engaging cascade via network $\boldsymbol{\Theta}$. Then the diffusion pathway of the cascade $c$ can be represented as

$$\boldsymbol{\Theta}^c \in \mathbb{R}_+^{N \times N} = \left(Z_1^c \boldsymbol{\Theta}_{\cdot 1} + (1 - Z_1^c)\boldsymbol{\Psi}_{\cdot 1}, \cdots, Z_N^c \boldsymbol{\Theta}_{\cdot N} + (1 - Z_N^c)\boldsymbol{\Psi}_{\cdot N}\right), \ c = 1, \cdots, C. \quad (3)$$

Therefore, $\boldsymbol{\Theta}^c$ is column-wise mixture of $\boldsymbol{\Theta}$ and $\boldsymbol{\Psi}$, which can vary for different cascades. Different to conventional mixture model, the proposed method allows $Z_i^c$ to vary across different cascade samples and nodes. Therefore, the proposed model (3) can generate up to $2^N$ different types of diffusion patterns, which grows exponentially as network size increases. Therefore, the proposed method has the principled heterogeneity modeling for the diffusion patterns. The likelihood of the proposed double mixture model can be also explicitly formulated. Following $\boldsymbol{P}_I$ and $\boldsymbol{P}_U$ in (1), we have the probability for node $i$ being infected in the $c$-th cascade given specific network as:

$$\boldsymbol{P}(t_i^c \mid Z_i^c; \{t_j^c : t_j^c < t_i^c\}, \boldsymbol{\Theta}^c) = \left[\boldsymbol{P}_I(t_i^c; \boldsymbol{\Theta}_{\cdot i})\right]^{Z_i^c} \times \left[\boldsymbol{P}_I(t_i^c; \boldsymbol{\Psi}_{\cdot i})\right]^{1 - Z_i^c}, \quad (4)$$

and following (2) the probability for node $i$ not being infected is

$$\boldsymbol{P}(T \mid Z_i^c; \{t_j^c : t_j^c < T\}, \boldsymbol{\Theta}^c) = \left[\boldsymbol{P}_U(T; \boldsymbol{\Theta}_{\cdot i})\right]^{Z_i^c} \times \left[\boldsymbol{P}_U(T; \boldsymbol{\Psi}_{\cdot i})\right]^{1 - Z_i^c}. \quad (5)$$

Denote $\boldsymbol{Z} = \{Z_i^c\}_{i=1, c=1}^{N, C}$, the joint distribution of diffusion indicators is

$$\boldsymbol{P}(\boldsymbol{Z}) = \prod_{c=1}^C \prod_{i=1}^N \boldsymbol{P}(Z_i^c) = \prod_{c=1}^C \prod_{i=1}^N \pi_i^{Z_i^c}(1 - \pi_i)^{1 - Z_i^c}. \quad (6)$$

Then we have the joint distribution of cascade samples and diffusion indicators $\boldsymbol{Z}$ as

$$\prod_{c=1}^C \boldsymbol{P}(\boldsymbol{t}^{(c)}, \boldsymbol{Z}; \boldsymbol{\Omega}) = \prod_{c=1}^C \left\{ \prod_{i:t_i^c \leqslant T} \left[\pi_i \boldsymbol{P}_I(t_i^c; \boldsymbol{\Theta}_{\cdot i})\right]^{Z_i^c} \left[(1 - \pi_i)\boldsymbol{P}_I(t_i^c; \boldsymbol{\Psi}_{\cdot i})\right]^{1 - Z_i^c} \right. \quad (7)$$

$$\left. \times \prod_{j:t_j^c \geqslant T} \left[\pi_j \boldsymbol{P}_U(T; \boldsymbol{\Theta}_{\cdot i})\right]^{Z_i^c} \left[(1 - \pi_j)\boldsymbol{P}_U(T; \boldsymbol{\Psi}_{\cdot i})\right]^{1 - Z_i^c} \right\}, \quad (8)$$

where $\boldsymbol{\Omega} = (\boldsymbol{\pi}, \boldsymbol{\Theta}, \boldsymbol{\Psi})$ denote model parameters. And the marginal distribution of cascade samples is

$$\prod_{c=1}^C \boldsymbol{P}(\boldsymbol{t}^{(c)}; \boldsymbol{\Omega}) = \prod_{c=1}^C \left\{ \prod_{i:t_i^c \leqslant T} \left[\pi_i \boldsymbol{P}_I(t_i^c; \boldsymbol{\Theta}_{\cdot i}) + (1 - \pi_i)\boldsymbol{P}_I(t_i^c; \boldsymbol{\Psi}_{\cdot i})\right] \times \right. \quad (9)$$

$$\left. \prod_{j:t_j^c \geqslant T} \left[\pi_i \boldsymbol{P}_U(T; \boldsymbol{\Theta}_{\cdot i}) + (1 - \pi_i)\boldsymbol{P}_U(T; \boldsymbol{\Psi}_{\cdot i})\right] \right\}.$$

Another advantage of our model is that the posterior distribution of diffusion indicator $\{Z_i^c\}$ can be calculated in an explicit form

$$\hat{\pi}_i^c := \boldsymbol{P}(Z_i^c = 1 \mid \boldsymbol{t}^{(c)}) = \begin{cases} \frac{\pi_i \boldsymbol{P}_I(t_i^c; \boldsymbol{\Theta}_{\cdot i})}{\pi_i \boldsymbol{P}_I(t_i^c; \boldsymbol{\Theta}_{\cdot i}) + (1 - \pi_i)\boldsymbol{P}_I(t_i^c; \boldsymbol{\Psi}_{\cdot i})}, & \text{if } t_i^c < T \\ \frac{\pi_i \boldsymbol{P}_U(t_i^c; \boldsymbol{\Theta}_{\cdot i})}{\pi_i \boldsymbol{P}_U(t_i^c; \boldsymbol{\Theta}_{\cdot i}) + (1 - \pi_i)\boldsymbol{P}_U(t_i^c; \boldsymbol{\Psi}_{\cdot i})}, & \text{if } t_i^c \geqslant T \end{cases} \quad (10)$$

### 3.3 Model identification

In this subsection, we establish the identifiability of the double network mixture model. Denote $\mathcal{R}_i^\Theta := \{j \in \{1, \cdots, N\} \mid \Theta_{ji} > 0\}$ and $\mathcal{R}_i^\Psi := \{j \in \{1, \cdots, N\} \mid \Psi_{ji} > 0\}$ as the sets of nodes that can directly reach node $i$ on $\Theta$ and $\Psi$, and $\boldsymbol{t}_{\mathcal{R}_i} \in [0, T]^{|\mathcal{R}_i^\Theta \cup \mathcal{R}_i^\Psi|} := \{t_j \mid j \in \mathcal{R}_i^\Theta \cup \mathcal{R}_i^\Psi\}$ as the infectious times of these nodes. We have the following identifiability result for model in (9).

**Proposition 1.** *For each node $i$, $i = 1, \cdots, N$, assume that 1) $\|\Theta_{\cdot i}\|_1 + \|\Psi_{\cdot i}\|_1 > 0$ and there exists $j \neq i$ such that $\Theta_{ji} \neq \Psi_{ji}$, and 2) survival function satisfies $S(t_i \mid t_j, \lambda_{ji}) = \exp\{\lambda_{ji} h(t_i - t_j)\}$ for some differentiable function $h(\cdot)$. Then, the parameters $(\pi_i, \Theta_{\cdot i}, \Psi_{\cdot i})$ associated with node $i$ in (9) are identifiable, i.e., if*

$$\pi_i \boldsymbol{P}_I(t; \Theta_{\cdot i}) + (1 - \pi_i) \boldsymbol{P}_I(t; \Psi_{\cdot i}) = \tilde{\pi}_i \boldsymbol{P}_I(t; \tilde{\Theta}_{\cdot i}) + (1 - \tilde{\pi}_i) \boldsymbol{P}_I(t; \tilde{\Psi}_{\cdot i}), \text{ or}$$

$$\pi_i \boldsymbol{P}_U(t; \Theta_{\cdot i}) + (1 - \pi_i) \boldsymbol{P}_U(t; \Psi_{\cdot i}) = \tilde{\pi}_i \boldsymbol{P}_U(t; \tilde{\Theta}_{\cdot i}) + (1 - \tilde{\pi}_i) \boldsymbol{P}_U(t; \tilde{\Psi}_{\cdot i})$$

*for any $t_i$ and $\boldsymbol{t}_{\mathcal{R}_i}$, then $\pi_i = \tilde{\pi}_i$, $\Theta_{\cdot i} = \tilde{\Theta}_{\cdot i}$, and $\Psi_{\cdot i} = \tilde{\Psi}_{\cdot i}$.*

Assumption 2) can be satisfied by popular risk models including Exp model, Pow model, Ray model, and other additive risk models of information propagation, such as kernel hazard functions [10] and feature-enhanced hazard functions [36]. Proposition 1 shows that the network preferences $\{\pi_i\}_{i=1}^N$ are identifiable and diffusion networks $\Theta$ and $\Psi$ are column-wise identifiable. However, similar to the labeling non-identifiability issue in finite mixture model [23], $\Theta$ and $\Psi$ may still not be globally identifiable without structural constraints due to column permutation. Specifically, the data distribution does not change if we swap $\Theta_{\cdot i}$ and $\Psi_{\cdot i}$ in $\Theta$ and $\Psi$ with other columns fixed.

**Layer-specific network structure constraint** Motivated by real-world applications, one can interpret $\Theta$ as the diffusion pathways over an observed social network $\boldsymbol{A} \in \{0, 1\}^{N \times N}$. Therefore, we can impose support constraints on $\Theta$ as

$$\Theta_{ij} \geqslant 0 \text{ if } \boldsymbol{A}_{ij} = 1; \ \Theta_{ij} = 0 \text{ if } \boldsymbol{A}_{ij} = 0.$$

Due to the sparse nature of real-world social networks, the support constraint also implicitly imposes sparsity constraint on $\Theta$. On the other hand, one can interpret $\Psi$ as the diffusion pathways via latent social relations of individuals that are not captured by the social network $\boldsymbol{A}$. The magnitude of $\Psi_{ij}$ reflects the social distance between individual $i$ and $j$ in terms of their latent factors. It has been found that social distance typically has or can be approximated by low-rank structure [34, 26, 27], since high-dimensional social factors can always be embedded into a low dimensional latent space that preserves social distances [34]. Therefore, we impose low-rank structure on $\Psi$ as

$$\text{rank}(\Psi) \leqslant r, \ 1 \leqslant r << N.$$

Imposing the above support constraint and low-rank structure allows $\Theta$ and $\Psi$ to capture complementary diffusion patterns driven by different types of relations. In addition, the structure constraints solve the above column-wise permutation issue [6]. Specifically, we introduce the matrix subspace $\Lambda_1(\Theta) = \{N \in \mathbb{R}^{N \times N} \mid \text{support}(N) \subseteq \boldsymbol{A}\}$. We also perform SVD on $\Psi = \boldsymbol{U}\Sigma\boldsymbol{V}^\top$ where $\boldsymbol{U}, \boldsymbol{V} \in \mathbb{R}^{N \times r}$ and $r$ is the rank of $\Psi$. Then, we define another matrix subspace as $\Lambda_2(\Psi) = \{\boldsymbol{U}X^\top + Y\boldsymbol{V}^\top \mid X, Y \in \mathbb{R}^{N \times k}\}$. We have the following result:

**Proposition 2.** *Given that the assumptions in Proposition 1 hold, then $\Theta$ and $\Psi$ are identifiable if:*

$$\max_{N \in \Lambda_1(\Theta), \|N\|_\infty \leqslant 1} \|N\|_2 \times \max_{N \in \Lambda_2(\Psi), \|N\|_2 \leqslant 1} \|N\|_\infty < 1, \tag{11}$$

*where $\|\cdot\|_2$ and $\|\cdot\|_\infty$ denote matrix operation norm and largest element in magnitude.*

The first term in (23) controls the rank of $\Theta$ given a fixed sparsity level, where a larger value indicates a lower rank. The second term controls the sparsity level of $\Psi$ given a fixed rank, where a larger value indicates a lower sparsity level. Intuitively, networks $\Theta$ and $\Psi$ can be globally identified given that they are well-separated in terms of either rank or sparsity.

### 3.4 Model estimation

Combining the distribution of cascade samples in Section 3.2 and the network structure constraints in Section 3.3, we estimate the model parameters $\Omega$ via constrained likelihood maximization as

$$\arg\max_{\Omega = \{\Theta, \Psi, \pi\}} \frac{1}{C} \sum_{c=1}^C \log \boldsymbol{P}(\boldsymbol{t}^{(c)}; \Omega) \quad \text{s.t. } \Theta \odot (\boldsymbol{I} - \boldsymbol{A}) = \boldsymbol{0}, \ \text{rank}(\Psi) \leqslant r,$$

where $I$ is a $N$-by-$N$ matrix with all elements being 1. However, both the likelihood function and the rank regularization are difficult to directly optimize. Therefore, we maximize the evidence lower bound of the log-likelihood function and replace the low-rank penalty with its convex relaxation as nuclear norm $\|\cdot\|_\star$. The optimization problem can thus be reformulated as follow

$$\underset{q(\boldsymbol{Z}),\boldsymbol{\Omega}}{\arg\max}\boldsymbol{E}_{q(\boldsymbol{Z})}\big[\frac{1}{C}\sum_{c=1}^{C}\log\boldsymbol{P}(\boldsymbol{t}^{(c)},\boldsymbol{Z};\boldsymbol{\Omega})\big]-\big[\boldsymbol{E}_{q(\boldsymbol{Z})}\log\boldsymbol{q}(\boldsymbol{Z})\big] \text{ s.t. } \boldsymbol{\Theta}\odot(1-\boldsymbol{A})=\boldsymbol{0},\ \|\boldsymbol{\Psi}\|_\star\leqslant\rho,$$

where $\boldsymbol{E}_{q(\boldsymbol{Z})}$ is the expectation of $\boldsymbol{Z}$ over distribution $q(\boldsymbol{Z})$. The above objective function can be optimized via EM algorithm by iteratively updating $q(\boldsymbol{Z})$ and $\boldsymbol{\Omega}$. Specifically, with the estimated parameters $\boldsymbol{\Omega}^{(s)}$ from the $s$-th step:

$$\textbf{E-step:}q(\boldsymbol{Z};\boldsymbol{\Omega}^{(s)})=\prod_{c=1}^{C}\prod_{i=1}^{N}\boldsymbol{P}(Z_i^c\mid\boldsymbol{t}^{(c)};\boldsymbol{\Omega}^{(s)})$$

$$\textbf{M-step:}\boldsymbol{\Omega}^{(s+1)}=\underset{\boldsymbol{\Omega}}{\arg\max}\frac{1}{C}\sum_{c=1}^{C}\boldsymbol{E}_{q(\boldsymbol{Z};\boldsymbol{\Omega}^{(s)})}\big[\log\boldsymbol{P}(\boldsymbol{t}^{(c)},\boldsymbol{Z};\boldsymbol{\Omega})\big]\text{ s.t. }\boldsymbol{\Theta}\odot(\boldsymbol{I}-\boldsymbol{A})=0,\ \|\boldsymbol{\Psi}\|_\star\leqslant\rho.$$

In E-step, the posterior distribution of network indicators $\hat{\pi}_i^c = \boldsymbol{P}(Z_i^c \mid \boldsymbol{t}^{(c)};\boldsymbol{\Omega}^{(s)})$ can be explicitly updated via (10). In M-step, the objective function $\boldsymbol{Q}(\boldsymbol{\Omega} \mid \boldsymbol{\Omega}^{(s)}) := \frac{1}{C}\sum_{c=1}^{C}\boldsymbol{E}_{q(\boldsymbol{Z};\boldsymbol{\Omega}^{(s)})}\big[\log\boldsymbol{P}(\boldsymbol{t}^{(c)},\boldsymbol{Z};\boldsymbol{\Omega})\big]$ can be decomposed as $\boldsymbol{Q}(\boldsymbol{\Omega}\mid\boldsymbol{\Omega}^{(s)})=\boldsymbol{Q}_1(\boldsymbol{\Theta}\mid\boldsymbol{\Omega}^{(s)})+\boldsymbol{Q}_2(\boldsymbol{\Psi}\mid\boldsymbol{\Omega}^{(s)})+\boldsymbol{Q}_3(\boldsymbol{\pi}\mid\boldsymbol{\Omega}^{(s)})$. The arguments $\boldsymbol{\Theta}$, $\boldsymbol{\Psi}$, and $\boldsymbol{\pi}$ can thus be updated parallelly in M-step as

$$\text{M.1}:\boldsymbol{\Theta}^{(s+1)}=\underset{\boldsymbol{\Theta}}{\arg\max}\,\boldsymbol{Q}_1(\boldsymbol{\Theta}\mid\boldsymbol{\Omega}^{(s)})\text{ s.t. }\boldsymbol{\Theta}\odot(\boldsymbol{I}-\boldsymbol{A})=\boldsymbol{0} \tag{12}$$

$$\text{M.2}:\boldsymbol{\Psi}^{(s+1)}=\underset{\boldsymbol{\Psi}}{\arg\max}\,\boldsymbol{Q}_2(\boldsymbol{\Psi}\mid\boldsymbol{\Omega}^{(s)})\text{ s.t. }\|\boldsymbol{\Psi}\|_\star\leqslant\rho \tag{13}$$

$$\text{M.3}:\boldsymbol{\pi}^{(s+1)}=\underset{\boldsymbol{\pi}}{\arg\max}\,\boldsymbol{Q}_3(\boldsymbol{\pi}\mid\boldsymbol{\Omega}^{(s)}) \tag{14}$$

When the structural network $\boldsymbol{A}$ is not observed, one can impose $l_1$-norm penalty to pursue sparsity structure in $\boldsymbol{\Theta}$. Accordingly, the constraint in optimization (12) is replaced by $\|\boldsymbol{\Theta}\|_1 \leqslant s'$, where $s' > 0$ is the sparsity tuning parameter. The main advantage of the proposed relaxation is that the M-step becomes a series of convex optimization problems. Specifically, denote parameter spaces $\boldsymbol{\Xi}_{\boldsymbol{\Theta}}(s) := \{\boldsymbol{\Theta} \in [0,\beta_1]^{N\times N} \mid \boldsymbol{\Theta}\odot(\boldsymbol{I}-\boldsymbol{A})=\boldsymbol{0}\}$ with $s = \|\boldsymbol{A}\|_0$, $\boldsymbol{\Xi}_{\boldsymbol{\Psi}}(\rho) := \{\boldsymbol{\Psi} \in [0,\beta_2]^{N\times N} \mid \|\boldsymbol{\Psi}\|_\star \leqslant \rho\}$, and $\boldsymbol{\Xi}_{\boldsymbol{\pi}} := \{\boldsymbol{\pi} \in [\epsilon, 1-\epsilon]^N\}$, where $\beta_1, \beta_2$ are nonnegative constants and $0 < \epsilon < 0.5$. We have the following result:

**Theorem 3.1.** *The parameter spaces $\boldsymbol{\Xi}_{\boldsymbol{\Theta}}(s)$, $\boldsymbol{\Xi}_{\boldsymbol{\Psi}}(\rho)$, and $\boldsymbol{\Xi}_{\boldsymbol{\pi}}$ are convex sets for any $s > 0$, $\rho > 0$, and $\boldsymbol{Q}_3(\boldsymbol{\pi} \mid \boldsymbol{\Omega}^{(s)})$ is concave on $\boldsymbol{\pi}$. If the hazard function $H(t \mid t', \lambda)$ satisfies $\frac{\partial H^2(t|t',\lambda)}{\partial\lambda^2} = 0$ for $t \geqslant t'$, then $\boldsymbol{Q}_1(\boldsymbol{\Theta} \mid \boldsymbol{\Omega}^{(s)})$ and $\boldsymbol{Q}_2(\boldsymbol{\Psi} \mid \boldsymbol{\Omega}^{(s)})$ are concave on $\boldsymbol{\Theta}$ and $\boldsymbol{\Psi}$, respectively. Furthermore, if for any node $i$, the probabilities of being source node $\boldsymbol{P}(v) > 0$ for $v \in \mathcal{R}$ where $\mathcal{R}$ denotes the set of nodes from which $i$ is reachable via a directed path, then $\mathbb{E}_t[\boldsymbol{Q}_1(\boldsymbol{\Theta} \mid \boldsymbol{\Omega}^{(s)})]$ and $\mathbb{E}_t[\boldsymbol{Q}_2(\boldsymbol{\Psi} \mid \boldsymbol{\Omega}^{(s)})]$ are also strictly concave in terms of $\boldsymbol{\Theta}$ and $\boldsymbol{\Psi}$, respectively.*

Theorem 3.1 guarantees that the optimization in M-step has a unique and optimal solution. Combining the theorem with the convergence guarantee of EM algorithm for convex ancillary function $\boldsymbol{Q}(\boldsymbol{\Omega} \mid \boldsymbol{\Omega}^{(s)})$ [2, 38], the above likelihood maximizer $\hat{\boldsymbol{\Omega}}$ is guaranteed to converge to the true $\boldsymbol{\Omega}$. In addition, the convexity assumption on hazard function can be satisfied by popular risk models including Exp model, Pow model, Ray model, and other additive risk models. We summarize and provide details for the above optimization process in the Appendix.

## 4 Numerical experiments on synthetic cascading data

We investigate the performance of the proposed method in recovering the diffusion networks based on synthetic cascading data. The performance of global transmission rates estimation on a network $\boldsymbol{\Theta}$ is measured by normalized mean absolute error (MAE) as $\textbf{MAE}(\boldsymbol{\Theta}) = \sum_{i,j}|\hat{\boldsymbol{\Theta}}_{ij} - \boldsymbol{\Theta}_{ij}|/\boldsymbol{\Theta}_{ij}$. The

performance of network probability $\boldsymbol{\pi}$ estimation is measured as $\mathbf{MAE}(\boldsymbol{\pi}) = \sum_i |\hat{\pi}_i - \pi_i|/\pi_i$. In addition, we investigate the performance of recovering the structure of diffusion network. Specifically, given a diffusion network $\boldsymbol{\Theta}$ we consider three topology estimation metrics accuracy, precision, and recall as $\mathbf{Acc}(\boldsymbol{\Theta}) = \sum_{i,j} |\mathbf{I}(\hat{\boldsymbol{\Theta}}_{ij}) - \mathbf{I}(\boldsymbol{\Theta}_{ij})|/(\sum_{i,j} \mathbf{I}(\hat{\boldsymbol{\Theta}}_{ij}) + \sum_{i,j} \mathbf{I}(\boldsymbol{\Theta}_{ij}))$, $\mathbf{Pre}(\boldsymbol{\Theta}) = \sum_{i,j}(\mathbf{I}(\hat{\boldsymbol{\Theta}}_{ij}) \cdot \mathbf{I}(\boldsymbol{\Theta}_{ij}))/\sum_{i,j} \mathbf{I}(\hat{\boldsymbol{\Theta}}_{ij})$, and $\mathbf{Recall}(\boldsymbol{\Theta}) = \sum_{i,j}(\mathbf{I}(\hat{\boldsymbol{\Theta}}_{ij}) \cdot \mathbf{I}(\boldsymbol{\Theta}_{ij}))/\sum_{i,j} \mathbf{I}(\boldsymbol{\Theta}_{ij})$, where $\mathbf{I}(\alpha) = 1$ if $\alpha > 0$ and $\mathbf{I}(\alpha) = 0$ otherwise. In the following numerical experiments, we fix the size of diffusion networks at $N = 200$.

## 4.1 Benchmark comparison under different network topologies

We compare the proposed method with baseline methods including NetRate [31], MMRate [37], and ConNIe [29] on diffusion network recovery. We generate diffusion networks $\boldsymbol{\Theta}$ and $\boldsymbol{\Psi}$ to mimic different structure of real-world networks. Specifically, we consider the diffusion network $\boldsymbol{\Theta}$ and its support $\boldsymbol{A}$ as random network, network with community structure, and scale-free network. On the other hand, we fix the latent diffusion network to be both low-rank (rank 5) and sparse (edge density 0.05). Given $\boldsymbol{\Theta}$ and $\boldsymbol{\Psi}$, we generate $C = 2,000$ independent cascade samples based on double mixture model with Exp transmission model and observation window length being $T = 10$.

Table 1 shows the proposed method outperforms both NetRate and MMRate by achieving lower MAE of transmission rate estimation on diffusion network $\boldsymbol{\Theta}$ under three network topologies and lower MAE of corresponding latent diffusion network $\boldsymbol{\Psi}$. MAE comparison does not include ConNIe since it only estimates network topology. In addition, we compare the proposed method with baseline methods in terms of network topology recovery on both $\boldsymbol{\Psi}$ and joint network $\hat{\boldsymbol{\Theta}} \cup \hat{\boldsymbol{\Psi}}$ for a fair comparison since benchmark methods NetRate and ConNIe do not distinguish the different diffusion networks by design. Table 1 shows the proposed method achieves higher accuracy on topology recovery via both latent network $\hat{\boldsymbol{\Psi}}$ and joint network $\hat{\boldsymbol{\Theta}} \cup \hat{\boldsymbol{\Psi}}$ than NetRate, MMRate, and ConNIe under different underlying structures in $\boldsymbol{\Theta}$. It also shows that under different settings, our method significantly outperforms all other methods in precision while remains higher recall than MMRate and ConNIe, and achieves similar recall to NetRate. Since only MMRate differentiates diffusion networks and estimates probabilities of cascading diffusing over a specific network, we compare $\mathrm{MAE}(\boldsymbol{\pi})$ from the proposed method with that from MMRate. Table 1 shows that our method also outperforms MMRate in estimation of network selection probability under different network topologies.

Table 1: Diffusion network estimations from different methods under three $\boldsymbol{\Theta}$ topologies (standard deviation in parenthesis, best performance highlighted in blue).

| | | $MAE_{\boldsymbol{\Theta}}$ | $MAE_{\boldsymbol{\Psi}}$ | $MAE_{\boldsymbol{\pi}}$ | $Acc_{\boldsymbol{\Psi}}$ | $Pre_{\boldsymbol{\Psi}}$ | $Rec_{\boldsymbol{\Psi}}$ | $Acc_{\boldsymbol{\Theta} \cup \boldsymbol{\Psi}}$ | $Pre_{\boldsymbol{\Theta} \cup \boldsymbol{\Psi}}$ | $Rec_{\boldsymbol{\Theta} \cup \boldsymbol{\Psi}}$ |
|---|---|---|---|---|---|---|---|---|---|---|
| | Rand | 0.315(0.011) | 0.544(0.007) | 0.047(0.002) | 0.842(0.006) | 0.757(0.008) | 0.948(0.006) | 0.889(0.006) | 0.829(0.006) | 0.958(0.004) |
| OurAlg | Com | 0.264(0.005) | 0.507(0.004) | 0.049(0.002) | 0.846(0.005) | 0.767(0.007) | 0.942(0.004) | 0.888(0.004) | 0.832(0.006) | 0.954(0.003) |
| | Scale | 0.312(0.006) | 0.545(0.006) | 0.050(0.002) | 0.833(0.005) | 0.750(0.007) | 0.937(0.005) | 0.883(0.004) | 0.826(0.007) | 0.949(0.003) |
| | Rand | 0.865(0.001) | 2.998(0.026) | – | 0.172(0.001) | 0.095(0.001) | 0.964(0.005) | 0.207(0.003) | 0.116(0.002) | 0.942(0.004) |
| NetRate | Com | 0.851(0.000) | 3.103(0.014) | – | 0.180(0.002) | 0.099(0.001) | 0.975(0.004) | 0.216(0.002) | 0.122(0.001) | 0.955(0.003) |
| | Scale | 0.855(0.000) | 2.897(0.018) | – | 0.179(0.002) | 0.099(0.001) | 0.929(0.003) | 0.220(0.002) | 0.125(0.001) | 0.922(0.003) |
| | Rand | 0.637(0.007) | 1.490(0.030) | 0.888(0.133) | 0.201(0.023) | 0.120(0.015) | 0.632(0.023) | 0.265(0.028) | 0.164(0.019) | 0.695(0.019) |
| MMRate | Com | 0.630(0.008) | 1.665(0.036) | 0.881(0.137) | 0.219(0.017) | 0.130(0.012) | 0.705(0.008) | 0.281(0.020) | 0.173(0.015) | 0.755(0.006) |
| | Scale | 0.668(0.006) | 1.477(0.027) | 0.928(0.028) | 0.211(0.019) | 0.126(0.013) | 0.650(0.009) | 0.278(0.023) | 0.173(0.018) | 0.710(0.007) |
| | Rand | – | – | – | 0.519(0.006) | 0.397(0.005) | 0.752(0.010) | 0.638(0.006) | 0.530(0.006) | 0.800(0.008) |
| ConNIe | Com | – | – | – | 0.519(0.005) | 0.398(0.005) | 0.748(0.007) | 0.638(0.005) | 0.532(0.007) | 0.796(0.006) |
| | Scale | – | – | – | 0.523(0.006) | 0.402(0.005) | 0.748(0.009) | 0.646(0.006) | 0.543(0.006) | 0.797(0.008) |

## 4.2 Network recovery under different transmission models

In this subsection, we investigate the performance of our diffusion network estimation when cascade samples are generated from popular transmission models including Exp, Pow, and Ray models, respectively. Table 2 illustrates the transmission rates recovery and network topology recovery on $\boldsymbol{\Psi}$ under different transmission models and cascade sample sizes $C$. As the sample size $C$ increases, both the parameter estimations (MAE) and topology recovery metrics (Acc, Pre, Rec) improve under different transmission models. In addition, the degree of improvement decreases as more cascade samples become available. This pattern indicates the consistency of the proposed diffusion network estimators, and the convergence of the proposed EM-type optimization. Notice that the diffusion network recovery based on cascade samples generated from Pow transmission model is better than

Exp and Ray model. This is because the Pow model introduces an additional parameter as time lag lower bound, which lowers the variation of activate time and overestimation of transmission rates.

Table 2: Diffusion network estimations from the proposed method under different cascade models and sample sizes.

|  |  | $MAE_{\Theta}$ | $MAE_{\Psi}$ | $MAE_{\pi}$ | $Acc_{\Psi}$ | $Pre_{\Psi}$ | $Rec_{\Psi}$ |
|---|---|---|---|---|---|---|---|
| Exp | C = 500 | 0.320(0.019) | 0.871(0.019) | 0.095(0.006) | 0.564(0.010) | 0.431(0.011) | 0.817(0.010) |
|  | C = 1000 | 0.235(0.013) | 0.667(0.007) | 0.091(0.005) | 0.743(0.009) | 0.653(0.014) | 0.862(0.007) |
|  | C = 1500 | 0.216(0.012) | 0.601(0.008) | 0.085(0.004) | 0.817(0.006) | 0.746(0.008) | 0.903(0.006) |
|  | C = 2000 | 0.187(0.011) | 0.562(0.007) | 0.082(0.004) | 0.857(0.004) | 0.794(0.006) | 0.930(0.005) |
| Ray | C = 500 | 0.287(0.018) | 1.053(0.056) | 0.152(0.008) | 0.605(0.018) | 0.525(0.024) | 0.714(0.013) |
|  | C = 1000 | 0.211(0.015) | 0.990(0.100) | 0.130(0.008) | 0.712(0.019) | 0.605(0.026) | 0.866(0.012) |
|  | C = 1500 | 0.179(0.011) | 0.629(0.091) | 0.134(0.004) | 0.795(0.017) | 0.702(0.026) | 0.916(0.007) |
|  | C = 2000 | 0.188(0.012) | 0.605(0.081) | 0.137(0.004) | 0.807(0.011) | 0.709(0.019) | 0.937(0.011) |
| Pow | C = 500 | 0.171(0.008) | 0.433(0.012) | 0.144(0.010) | 0.882(0.015) | 0.832(0.024) | 0.939(0.009) |
|  | C = 1000 | 0.129(0.006) | 0.328(0.007) | 0.131(0.007) | 0.956(0.004) | 0.932(0.007) | 0.982(0.004) |
|  | C = 1500 | 0.112(0.005) | 0.290(0.004) | 0.125(0.008) | 0.969(0.002) | 0.946(0.004) | 0.993(0.002) |
|  | C = 2000 | 0.106(0.006) | 0.271(0.004) | 0.129(0.011) | 0.973(0.003) | 0.950(0.005) | 0.997(0.001) |

## 5  Real cascading data analysis

In this section, we study the cascading patterns of research topics in sociology by discovering the diffusion networks among US universities. Geographic proximity is known to facilitate idea exchanges through collaborations and citations among colleagues within the same institution or nearby locations [3], which can be due to dependence on shared research resources and the need for coordination [28]. It is also known that research diffusion can happen via the latent network of scholars, which is also sometimes called the "invisible colleges" [7] to highlight the role of informal networks and latent ties. Our goal is to infer the latent research topic diffusion network and compare its difference with the geographic network in terms of topic cascading.

**Data preparation and preprocess** We select universities in both the list of 1965 ASA Guide to Graduate Departments of Sociology and the list of 2022 US News Best Sociology Programs in America. In addition, we exclude the universities whose sociology programs were established after 1965. Based on the above conditions, we finalize $N = 104$ universities, which are considered as the target population of our study. We then create a geographical network $\boldsymbol{A} \in \{0,1\}^{N \times N}$ among the selected universities, where $\boldsymbol{A}_{ij} = 1$ if university $i$ and $j$ are located in the same state.

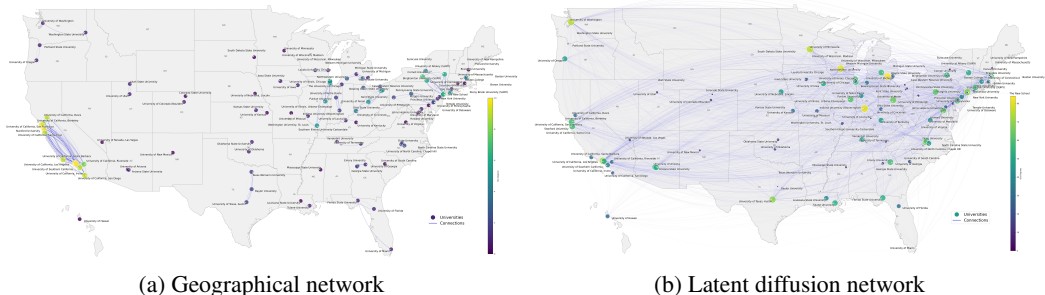

| (a) Geographical network | (b) Latent diffusion network |
|---|---|

Figure 2: Cascade transmission rates over geographical network $\boldsymbol{\Theta}$ and latent diffusion network $\boldsymbol{\Psi}$.

To construct research topics, we use Elsevier's Scopus API to compile a dataset of 29,725 unique articles from 23 top generalist sociology journals, including information on authors, their affiliations, article titles, and abstracts. Multiple keywords are extracted from the titles and abstracts of articles. After these preprocesses, we obtain 3,033 unique research topics that cover major research fields in sociology. For each research topic $c$, we construct the corresponding cascade samples as $(t_1^c, t_2^c, \cdots, t_N^c)$ where $t_i^c := \tilde{t}_i^c - \tilde{t}_0^c$, with $\tilde{t}_0^c$ being the publication date of the first article that involves topic $c$, and $\tilde{t}_i^c$ being the publication date of the first article that involves topic $c$ and is published by any scholar affiliated with university $i$. If university $i$ never publishes any article that involves topic $c$, we set $t_i^c$ to be year 2022.

**Diffusion networks inference** We simultaneously estimate the geographic diffusion network $\boldsymbol{\Theta}$ and the latent diffusion network $\boldsymbol{\Psi}$ based on the proposed model. Figure (2) illustrates the inferred diffu-

sion networks $\hat{\Theta}$ and $\hat{\Psi}$, respectively. The size of nodes represents the out-degree of corresponding universities on networks. We see that the geographic diffusion network has a strong local community structure, especially among universities in California, the Great Lakes, and the northeast coast.

Figure (2) also shows that the latent diffusion network demonstrates a decentralized structure and contains many connections between the east and west coasts. This captures the national collaboration and academic mobility. We also compare estimated pairwise transmission rates on diffusion networks $\hat{\Theta}$ and $\hat{\Psi}$ in Figure (3). In summary, the latent diffusion network is denser than the geographic network, and the magnitude of transmission rates on the geographic network is larger and has more variation compared to the latent diffusion network.

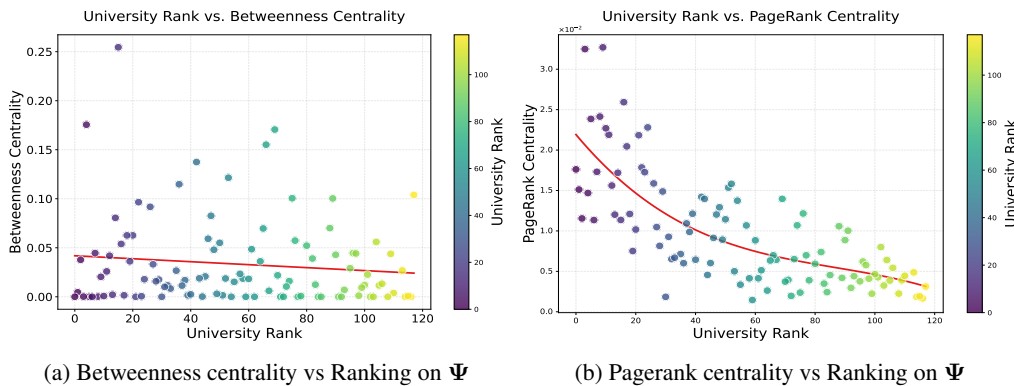

Figure 3: Comparison of distributions of pairwise transmission rates from geographic network $\Theta_{ij}$ and latent network $\Psi_{ij}$.

To provide an interpretation of the inferred geographic diffusion network $\Theta$ and latent diffusion network $\Psi$, we connect universities' positions on the network with universities' U.S. News University rankings on the sociology program. We choose these rankings since they are a systematic and popular summary of academic factors. Figure (4a) illustrates the association between program ranking and the node-wise betweenness centrality on latent diffusion network. The Betweenness centrality of a node $i$ is defined as $\sum_{j \neq i \neq k} \sigma_{jk}(i)/\sigma_{jk}$, where $\sigma_{jk}$ is the total number of shortest paths from node $j$ to $k$, and $\sigma_{jk}(i)$ is the number of those paths that pass through $i$. Based on the figure, the universities'

| (a) Betweenness centrality vs Ranking on $\Psi$ | (b) Pagerank centrality vs Ranking on $\Psi$ |
| --- | --- |

Figure 4: Node-wise centrality vs Ranking on latent network $\Psi$.

betweenness centralities are relatively uniform and do not significantly decrease as ranking increases. This suggests that there does not exist any strong community or centralized topology in the latent network. When universities are grouped according to their ranking (top 20, middle 60, low 20 etc.), each group has different high-betweenness universities, serving as the bridges in idea diffusion. We also investigate the relation between university rankings and Pagerank centrality. Pagerank centrality measures the influence of a node on a network based on how many influential nodes it connects to. Figure (4b) shows that the universities' Pagerank centralities decrease as their rankings increase, which suggests that the idea exchanges among high-ranked universities are more frequent and faster than those of other universities. This pattern appears because high-ranked universities have a higher level of research activities. University prestige is also considered as an important proxy for the quality of ideas, which increases the likelihood of research ideas from higher-ranked universities to diffuse.

## 6 Scalability analysis

The implicit convex nature of our objective function allows the optimization process to be as efficient as gradient descent, even though the optimization of our method is based on EM-type update. The E-step in our algorithm has an explicit form as in (10). Furthermore, we have proved in Theorem

3.1 that the objective function $Q(\mathbf{\Omega})$ in the M-step is strictly concave and in each EM-iteration, we only perform gradient ascent once instead of maximizing $Q(\mathbf{\Omega})$ until convergence. Under these conditions, established proofs in [2] show that the above EM-update is equivalent to gradient decent, and our optimization thus enjoys gradient descent's geometric convergence rate.

Computational complexity of the proposed method is $K \cdot O(N^2 \cdot C) + K \cdot O(N^3)$, where $N$ is network size, $C$ is sample size, and $K$ is the number of iterations executed. In practice, we require $C \gg N$ for reasonable estimation. The first term thus dominates and the computational complexity is approximately the same as that of gradient descent. We can replace the standard SVD operation in the $M$-step by randomized SVD or Lanczos algorithm. Then, we can further reduce the computational complexity of the second term from $\mathcal{O}(N^3)$ to $\mathcal{O}(rN^2)$ or $\mathcal{O}((r+l)N^2)$ where $r \ll N$ is the rank of network $\mathbf{\Psi}$, $N$ is the size of network, and $l$ is a constant usually between 10 and 20.

We also numerically investigate both time and estimation performance of the proposed method in recovering diffusion networks based on synthetic cascading data in large network settings. We adopt the same criteria used in Section 4 to evaluate estimation performance and use average time per iteration to evaluate time performance. We generate diffusion networks $\mathbf{\Theta}$ and $\mathbf{\Psi}$ of different sizes in a similar way to Section 4.2, fixing $\mathbf{\Psi}$ to be both low-rank (rank 5) and sparse (edge density 0.01). Given $\mathbf{\Theta}$ and $\mathbf{\Psi}$, we generate $C = 50,000$ independent cascade samples based on double mixture model with Exp transmission model and observation window length being $T = 10$.

Table 3: Execution time of different methods under different network sizes (unit: second per iteration).

|  | $N = 500$ | $N = 1000$ | $N = 2000$ | $N = 4000$ |
|---|---|---|---|---|
| OurAlg | 2.395(0.001) | 8.201(0.019) | 33.442(0.031) | 132.955(0.046) |
| NetRate | 1.390(0.001) | 4.929(0.011) | 20.039(0.030) | 79.992(0.040) |

Table 3 shows that the computation time of the proposed method per iteration is proportional to the squared network size $N^2$. This aligns with the earlier theoretical analysis of computational complexity. In addition, for all network sizes $N = 500, 1000, 2000, 4000$, the ratio between computational time per iteration of the proposed method and NetRate [31]is approximately a constant 1.65. Since NetRate [31] is well-known for its scalability, this result demonstrates the computational efficiency and scalability of the proposed method.

Table 4 illustrates the estimation performance of the proposed method for large networks. For networks of all sizes $N = 500, 1000, 2000$, the proposed method outperforms NetRate [31] on both transmission rate estimation and network topology recovery. For networks of sizes $N = 500, 1000$, the proposed method achieves estimation performances comparable to those of smaller networks in Section 4. Additionally, when sample size is fixed, the performance of the proposed method decreases, indicating the need for more samples on larger networks to achieve accurate estimations.

Table 4: Diffusion network estimations from the proposed method under different network sizes.

|  |  | $MAE_{\mathbf{\Theta}}$ | $MAE_{\mathbf{\Psi}}$ | $Acc_{\mathbf{\Psi}}$ | $Pre_{\mathbf{\Psi}}$ | $Rec_{\mathbf{\Psi}}$ |
|---|---|---|---|---|---|---|
|  | N = 500 | 0.398(0.001) | 0.711(0.019) | 0.894(0.003) | 0.820(0.005) | 0.983(0.001) |
| OurAlg | N = 1000 | 0.411(0.005) | 0.797(0.007) | 0.779(0.001) | 0.666(0.001) | 0.939(0.003) |
|  | N = 2000 | 0.437(0.003) | 0.814(0.008) | 0.571(0.002) | 0.523(0.002) | 0.630(0.002) |
|  | N = 500 | 0.743(0.014) | 1.053(0.038) | 0.078(0.008) | 0.041(0.004) | 0.818(0.007) |
| NetRate | N = 1000 | 0.755(0.010) | 1.094(0.019) | 0.086(0.001) | 0.045(0.001) | 0.914(0.007) |
|  | N = 2000 | 0.759(0.003) | 1.096(0.002) | 0.083(0.000) | 0.044(0.000) | 0.913(0.000) |

# 7 Conclusion

In this paper, we propose a novel double network mixture model for heterogeneous cascading process on multi-layer networks. Our method can identify the latent diffusion network complementary to the observed network. Due to its convex formulation, our method has both statistical and computational guarantee in terms of estimating diffusion networks. A major future work is to generalize the mixture graph model to a system with more than two-layer networks and derive the model identification conditions. Extending the proposed method to inference on time-varying networks with time-dependent parameters $\{\mathbf{\Theta}(t), \mathbf{\Psi}(t), \boldsymbol{\pi}(t)\}$ or nonparametric transmission models are also interesting directions for future works.

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

# A  Additional details on methodology

## A.1  Detailed expression of objectives in M-step

$Q_1(\boldsymbol{\Theta} \mid \boldsymbol{\Omega}^{(s)})$, $Q_2(\boldsymbol{\Psi} \mid \boldsymbol{\Omega}^{(s)})$, and $Q_3(\boldsymbol{\pi} \mid \boldsymbol{\Omega}^{(s)})$ can be formulated in detail as:

$$Q_1(\boldsymbol{\Theta} \mid \boldsymbol{\Omega}^{(s)}) = \frac{1}{C} \sum_{c=1}^{C} \Big\{ \sum_{i:t_i^c \leqslant T} \hat{\pi}_i^c \log \boldsymbol{P}_I(t_i; \boldsymbol{\Theta}_{\cdot i}) + \sum_{j:t_j^c > T} \hat{\pi}_j^c \log \boldsymbol{P}_U(T; \boldsymbol{\Theta}_{\cdot j}) \Big\}$$

$$Q_2(\boldsymbol{\Psi} \mid \boldsymbol{\Omega}^{(s)}) = \frac{1}{C} \sum_{c=1}^{C} \Big\{ \sum_{i:t_i^c \leqslant T} (1 - \hat{\pi}_i^c) \log \boldsymbol{P}_I(t_i; \boldsymbol{\Psi}_{\cdot i}) + \sum_{j:t_j^c > T} (1 - \hat{\pi}_j^c) \log \boldsymbol{P}_U(T; \boldsymbol{\Psi}_{\cdot j}) \Big\}$$

$$Q_3(\boldsymbol{\pi} \mid \boldsymbol{\Omega}^{(s)}) = \frac{1}{C} \sum_{c=1}^{C} \Big\{ \sum_{i=1}^{N} \Big[ \hat{\pi}_i^c \log \pi_i + (1 - \hat{\pi}_i^c) \log(1 - \pi_i) \Big] \Big\},$$

where $\boldsymbol{P_I}$, $\boldsymbol{P_U}$ follow the definitions in Section 3.1.

## A.2  Optimization algorithm and discussions

We summarize the optimization of the reformulated problem in Section 3.4 via EM algorithm as Algorithm 1.

---

**Algorithm 1** First-order projected EM algorithm

---

**Require:** initialization $\boldsymbol{\Omega}^{(0)} = \{\boldsymbol{\Theta}^{(0)}, \boldsymbol{\Psi}^{(0)}, \boldsymbol{\pi}^{(0)}\}$, observed network $\boldsymbol{A}$, low-rank penalty $\mu$, learning rate $\lambda$, and stopping criterion $\epsilon$.
  **while** $Q(\boldsymbol{\Omega}^{(s+1)} \mid \boldsymbol{\Omega}^{(s)}) - \mu\|\boldsymbol{\Psi}^{(s+1)}\|_\star - Q(\boldsymbol{\Omega}^{(s)} \mid \boldsymbol{\Omega}^{(s)}) + \mu\|\boldsymbol{\Psi}^{(s)}\|_\star > \epsilon$ **do**.
    **E-step**: update $\hat{\pi}_i^c = \boldsymbol{P}(Z_i \mid \boldsymbol{t}^c; \boldsymbol{\Omega}^{(s)})$ via its posterior distribution based on $\boldsymbol{\Omega}^{(s)}$ for $i = 1, \cdots, N$, $c = 1, \cdots, C$.
    **M-step**: decompose $\boldsymbol{Q}(\boldsymbol{\Omega} \mid \boldsymbol{\Omega}^{(s)}) = \boldsymbol{Q_1}(\boldsymbol{\Theta} \mid \boldsymbol{\Omega}^{(s)}) + \boldsymbol{Q_2}(\boldsymbol{\Psi} \mid \boldsymbol{\Omega}^{(s)}) + \boldsymbol{Q_3}(\boldsymbol{\pi} \mid \boldsymbol{\Omega}^{(s)})$
    M.1: Update $\boldsymbol{\Theta}$ via $\boldsymbol{Q_1}(\boldsymbol{\Theta} \mid \boldsymbol{\Omega}^{(s)})$:
        $\boldsymbol{\Theta}^{(s+1)} \leftarrow \max\{\boldsymbol{\Theta}^{(s)} + \lambda\frac{\partial \boldsymbol{Q_1}}{\boldsymbol{\Theta}}\big|_{\boldsymbol{\Theta}=\boldsymbol{\Theta}^{(s)}} \odot \boldsymbol{A}, \boldsymbol{0}\}$
    M.2: Update $\boldsymbol{\Psi}$ via $\boldsymbol{Q_2}(\boldsymbol{\Psi} \mid \boldsymbol{\Omega}^{(s)})$:
        M.2.1: $\boldsymbol{\Psi} \leftarrow \boldsymbol{\Psi} + \lambda\frac{\partial \boldsymbol{Q_2}}{\boldsymbol{\Psi}}\big|_{\boldsymbol{\Psi}=\boldsymbol{\Psi}^{(s)}}$
        M.2.2: perform SVD decomposition on $\boldsymbol{\Psi} = \boldsymbol{U}\boldsymbol{\Lambda}\boldsymbol{V}^\top$
        M.2.3: $\boldsymbol{\Psi}^{(s+1)} \leftarrow \max\{\boldsymbol{U}\mathrm{diag}(\boldsymbol{\Lambda} - \lambda\mu)_+\boldsymbol{V}, \boldsymbol{0}\}$
    M.3: Update $\boldsymbol{\pi}$ via $\boldsymbol{Q_3}(\boldsymbol{\pi} \mid \boldsymbol{\Omega}^{(s)})$:
        $\boldsymbol{\pi}_i^{(s+1)} = \frac{\sum_{c=1}^{C} \hat{\pi}_i^c}{C}$, $i = 1, \cdots, N$
**end while**

---

In Algorithm 1, we utilize projected gradient ascent and proximal gradient ascent to update diffusion networks $\boldsymbol{\Theta}$ and $\boldsymbol{\Psi}$ in M.1 and M.2 of M-step, where the latter can be implemented via singular value soft-thresholding operation [14, 5]. For computational efficiency and stability, we utilize the first-order EM algorithm [2] such that the ELBO is increased via one-step gradient ascend instead of maximized in M-step. In addition, both the gradients $\frac{\partial \boldsymbol{Q_1}}{\boldsymbol{\Theta}}$ and $\frac{\partial \boldsymbol{Q_2}}{\boldsymbol{\Psi}}$ have closed forms and can be efficiently calculated. The closed form gradients are provided in A.3.

**Likelihood-based parameter tuning** The low-rank penalty $\mu$ in the above Algorithm 1 can be selected in a data-adapted manner. Specifically, we can first randomly separate the total cascade samples into a training subset $C_{train}$ and a validation subset $C_{val}$, and estimate model parameters $\hat{\boldsymbol{\Omega}}_\mu$ on $C_{train}$ given a specific $\mu$. Then we calculate the log-likelihood of validation samples as $\frac{1}{|C_{val}|} \sum_{c \in C_{val}} \log \boldsymbol{P}(\boldsymbol{t}^{(c)}, \hat{\boldsymbol{\Omega}}_\mu)$, and select $\mu$ such that

$$\mu = \arg\max_{\mu \in \boldsymbol{G}} \frac{1}{|C_{val}|} \sum_{c \in C_{val}} \log \boldsymbol{P}(\boldsymbol{t}^{(c)}, \hat{\boldsymbol{\Omega}}_\mu)$$

where $G$ is a grid of candidates for penalty values.

**Sparsity pursuit on $\Theta$** When the observed network $A$ is not sparse enough or the sample size of cascade is relatively small, one can further add $l_1$ regularizer on $\Theta$ in the objective function $Q_1(\Theta \mid \Omega^{(s)})$. Accordingly, the matrix $\Theta$ is updated via an additional soft-thresholding operator in the M.1 step of Algorithm 1. Previous results shows that adopting the $l_1$ regularization can improve the performance and efficiency to recover the diffusion network structure [8, 31].

### A.3 Closed form gradients in Algorithm 1

Denote $(S_c^{\Theta})_{ji} = S(t_i^c \mid t_j^c, \Theta_{ji})$, $(S_c^{\Psi})_{ji} = S(t_i^c \mid t_j^c, \Psi_{ji})$, $(H_c^{\Theta})_{ji} = H(t_i^c \mid t_j^c, \Theta_{ji})$, and $(H_c^{\Psi})_{ji} = H(t_i^c \mid t_j^c, \Psi_{ji})$, where $S(\cdot \mid \cdot, \cdot)$ and $H(\cdot \mid \cdot, \cdot)$ are survival and hazard functions, respectively.

In addition, we introduce two cascade sample indicator variables $\mathbb{I}^{(1)}(c)_{ji}$ and $\mathbb{I}^{(2)}(c)_{ji}$ for each node pair $(j, i)$ such that

$$\mathbb{I}^{(1)}(c)_{ji} = \begin{cases} 1, & \text{if } t_j^c < t_i^c, \text{and}, t_j^c \leqslant T, \\ 0, & \text{otherwise} \end{cases}$$

and

$$\mathbb{I}^{(2)}(c)_{ji} = \begin{cases} 1, & \text{if } t_j^c < t_i^c, \ t_j^c \leqslant T, \ \text{and}, t_i^c \leqslant T, \\ 0, & \text{otherwise} \end{cases}$$

Then, $\frac{\partial Q_1}{\Theta}$ and $\frac{\partial Q_2}{\Psi}$ over each element in $\Theta$ and $\Psi$, respectively, can be explicitly formulated as:

$$\frac{\partial Q_1}{\partial (\Theta)_{ji}} = \sum_{c=1}^{C} \Big[ \mathbb{I}^{(1)}(c)_{ji} \times \mathbb{I}(A_{ji} = 1) \frac{\hat{\pi}_i^c}{(S_c^{\Theta})_{ji}} \frac{\partial (S_c^{\Theta})_{ji}}{\partial \Theta_{ji}} +$$

$$\mathbb{I}^{(2)}(c)_{ji} \times \mathbb{I}(A_{ji} = 1) \frac{\hat{\pi}_i^c}{\sum_{k:t_k^c < t_i^c}(H_c^{\Theta})_{ki}} \frac{\partial (H_c^{\Theta})_{ji}}{\partial \Theta_{ji}} \Big],$$

$$\frac{\partial Q_2}{\partial (\Psi)_{ij}} = \sum_{c=1}^{C} \Big[ \mathbb{I}^{(1)}(c)_{ji} \frac{1 - \hat{\pi}_i^c}{(S_c^{\Psi})_{ji}} \frac{\partial (S_c^{\Psi})_{ji}}{\partial \Psi_{ji}} + \mathbb{I}^{(2)}(c)_{ji} \frac{1 - \hat{\pi}_i^c}{\sum_{k:t_k^c < t_i^c}(H_c^{\Psi})_{ki}} \frac{\partial (H_c^{\Psi})_{ji}}{\partial \Psi_{ji}} \Big]$$

The above gradients can be calculated by matrix operations after translating two indicator functions into matrices.

## B  Additional details on experimental setup

### B.1  Benchmark comparison under different network topologies

In this experiment, we generate diffusion networks with different topologies. Specifically, we consider the diffusion network $\Theta$ and its support $A$ as random network, network with community structure, and scale-free network. For random network, we generate $A$ via Erdos-Renyi model with edge generation probability being $0.01$. For the community structure, we generate $A$ with a stochastic block model that contains four equal-sized communities. The generation probability of within-community edge is $0.05$ and that of between-community edge is $0.01$. For scale-free network, we generate $A$ via Barabási–Albert model where we set the number of edges to attach from a new node to existing nodes to be 1. After generating support $A$, we set its diagonal to be 0 and assign to each non-zero edge a weight that follows $Unif(1, 5)$ to construct $\Theta$.

We generate the latent diffusion network as $\Psi = \Psi_1 \Psi_2^\top$ where $\Psi_1, \Psi_2 \in \mathbb{R}_+^{N \times 5}$. We fix the proportion of non-zero edges in $\Psi_1, \Psi_2$ at $0.1$ and sample weights from $Unif(1, 2)$ for these non-zero edges. The generated $\Psi$ has an edge density of $0.05$ and transmission rates between 1 and 8 on non-zero edges.

In addition, we impute edges on $A$ to ensure that each row and column of $\Theta \cup \Psi$ has at least one non-zero element, i.e., the combined network is connected. Given $\Theta$ and $\Psi$, we generate $C = 2,000$ independent cascade samples based on the double mixture model with Exp transmission model and observation window length of $T = 10$.

## B.2 Network recovery under different transmission models

In this experiment, we generate the latent diffusion network $\boldsymbol{\Psi}$ with low-rank structure using the same procedure as in B.1.

Given $\boldsymbol{\Psi}$ and its support $\boldsymbol{B}$, we then generate $\boldsymbol{\Theta}$'s support $\boldsymbol{A}$ by forcing one overlap with $\boldsymbol{B}$ per column, i.e. $\{(i,j) \mid \boldsymbol{A}_{ij}\boldsymbol{B}_{ij} = 1\}$, and one non-overlap with $\boldsymbol{B}$ per column, i.e. $\{(i,j) \mid \boldsymbol{A}_{ij} = 1, \boldsymbol{B}_{ij} = 0\}$. For Exp and Pow transmission models, we create $\boldsymbol{\Theta}$ by sampling transmission rates from $Unif(2,5)$ for non-zero edges of $\boldsymbol{A}$. We slightly modify the generation process for Ray model by decreasing the transmission rates on non-zero edges of $\boldsymbol{\Theta}$ and $\boldsymbol{\Psi}$. We instead sample weights from $Unif(0.1, 0.8)$ for non-zero edges in $\Psi_1, \Psi_2$ and weights from $Unif(0.02, 2)$ for those in $\boldsymbol{\Theta}$. This modification aims to force significantly large differences in infection time that contribute to large differences in probability of infection among potential parents.

We also add edges in $\boldsymbol{\Theta}$ to avoid all-zero row or column in $\boldsymbol{\Theta} \cup \boldsymbol{\Psi}$ to ensure diffusion network connectivity. The resulting $\boldsymbol{\Theta}$ and $\boldsymbol{\Psi}$ have edge density at 0.01 and 0.05, respectively. Based on $\boldsymbol{\Theta}$ and $\boldsymbol{\Psi}$, we generate different numbers of cascade samples at $C = 500, 1000, 1500, 2000$ with a fix observation window length at $T = 10$. In the Pow transmission model, we select delay parameter $\delta = 1$.

## B.3 Network recovery under large network settings

In this experiment, for each network size $N = 500, 1000, 2000, 4000$, we generate the latent diffusion network $\boldsymbol{\Psi}$ following the same procedure as in B.2 but fixing its rank at 5 and edge density at 0.01. Given $\boldsymbol{\Psi}$, we also follow the same procedure in B.2 to generate $\boldsymbol{\Theta}$. The resulting $\boldsymbol{\Theta}$ has an edge density of 0.001. Based on $\boldsymbol{\Theta}$ and $\boldsymbol{\Psi}$, we generate $C = 50000$ independent cascade samples with a fixed observation window length of $T = 10$.

To cope with memory limits when the network size $N$ and the sample size $C$ are large, in each outer iteration, we stream the data in batches of size $B$: we process one batch at a time to compute the contributions (likelihood terms, gradients) required by both the E-step and the M-step, accumulate these quantities across all $\lceil M/B \rceil$ batches, and then carry out the parameter update using the aggregated totals—thus matching the effect of a full-batch EM update while keeping memory usage bounded. When more memory is available, one can increase the batch size $B$ accordingly for better parallelization on GPU and thus shorter execution time per iteration.

# C Additional numerical results

## C.1 Network recovery under different support overlap

In this subsection, we investigate the performance of our proposed method under different levels of overlap between $\boldsymbol{\Theta}$ and $\boldsymbol{\Psi}$, defined as overlap$(\boldsymbol{\Theta}, \boldsymbol{\Psi}) = \sum_{i,j} \mathbf{I}(\boldsymbol{A}_{ij} \times \boldsymbol{B}_{ij}) / \sum_{i,j} \mathbf{I}(\boldsymbol{A}_{ij} + \boldsymbol{B}_{ij})$, where $\boldsymbol{A}, \boldsymbol{B}$ are the support of $\boldsymbol{\Theta}, \boldsymbol{\Psi}$, respectively.

We generate $\boldsymbol{\Psi} = \Psi_1 \Psi_2^T$ following similar procedure in B.1 except that we sample weights of non-zero edges in $\Psi_1$ and $\Psi_2$ from $Unif(0.2, 1.5)$ and increase its rank to 30 to better control its overlap degree with $\boldsymbol{\Theta}$. The edge density of the generated $\boldsymbol{\Psi}$ is 0.025 and the transmission rates on its non-zero edges range from 0.05 to 2.

Given $\boldsymbol{\Psi}$ and its support $\boldsymbol{B}$, we generate different supports $\boldsymbol{A}$ following the same procedure in B.2 while changing their overlaps and non-overlaps per column with $\boldsymbol{B}$ so that overlap$(\boldsymbol{\Theta}, \boldsymbol{\Psi})$ takes three levels at about $0.1, 0.3, 0.5$. Then we create $\boldsymbol{\Theta}$ by sampling transmission rates from $Unif(0.1, 0.2) \cup (1.9, 2.0)$ for the overlap $\{(i,j) \mid \boldsymbol{A}_{ij}\boldsymbol{B}_{ij} = 1\}$, and sampling from $Unif(1, 2)$ for non-overlap support $\{(i,j) \mid \boldsymbol{A}_{ij} = 1, \boldsymbol{B}_{ij} = 0\}$.

We also add edges in $\boldsymbol{\Theta}$ with transmission rates from $Unif(1, 2)$ to avoid all-zero row or column in $\boldsymbol{\Theta} \cup \boldsymbol{\Psi}$ to ensure diffusion network connectivity. Based on $\boldsymbol{\Theta}$ and $\boldsymbol{\Psi}$, we generate different numbers of cascade samples from Exp transmission model, and fix the observation window length at $T = 10$.

Table 5 shows that the diffusion network recovery performance of the proposed method improves as sample sizes increases. Additionally, as overlap level between $\boldsymbol{\Theta}$ and $\boldsymbol{\Psi}$ increases, the transmission rates estimations MAE on both $\boldsymbol{\Theta}, \boldsymbol{\Psi}$ and $\boldsymbol{\pi}$ decreases while estimation of $\boldsymbol{\Psi}$'s support improves.

This pattern uncovers a trade-off between network differentiation and topology recovery in network mixture model. As the overlap level increases, the joint diffusion complexity on $\Theta$ and $\Psi$ decreases due to the shared diffusion pathways and support information. On the other hand, it becomes more difficult and requires more samples to differentiate $\Theta$ and $\Psi$ on their overlap component.

Table 5: Diffusion network estimations from the proposed method under different degrees of overlapping between diffusion networks.

| overlap($\Theta, \Psi$) | | $MAE_{\Theta}$ | $MAE_{\Psi}$ | $MAE_{\pi}$ | $Acc_{\Psi}$ | $Pre_{\Psi}$ | $Rec_{\Psi}$ |
|---|---|---|---|---|---|---|---|
| 0.1 | C = 500 | 0.325(0.014) | 0.510(0.018) | 0.314(0.013) | 0.690(0.007) | 0.553(0.009) | 0.919(0.005) |
| | C = 1000 | 0.309(0.015) | 0.414(0.016) | 0.300(0.016) | 0.793(0.008) | 0.669(0.010) | 0.974(0.003) |
| | C = 1500 | 0.307(0.016) | 0.391(0.018) | 0.297(0.016) | 0.827(0.013) | 0.711(0.018) | 0.989(0.002) |
| | C = 2000 | 0.277(0.012) | 0.360(0.012) | 0.288(0.015) | 0.853(0.004) | 0.748(0.007) | 0.994(0.002) |
| 0.3 | C = 500 | 0.393(0.012) | 0.577(0.017) | 0.435(0.016) | 0.764(0.008) | 0.647(0.010) | 0.933(0.008) |
| | C = 1000 | 0.388(0.018) | 0.519(0.016) | 0.454(0.015) | 0.852(0.006) | 0.756(0.010) | 0.977(0.003) |
| | C = 1500 | 0.387(0.013) | 0.507(0.018) | 0.456(0.012) | 0.887(0.008) | 0.804(0.013) | 0.989(0.002) |
| | C = 2000 | 0.366(0.017) | 0.489(0.022) | 0.452(0.018) | 0.908(0.003) | 0.835(0.004) | 0.994(0.002) |
| 0.5 | C = 500 | 0.458(0.014) | 0.618(0.014) | 0.533(0.017) | 0.810(0.017) | 0.712(0.023) | 0.940(0.006) |
| | C = 1000 | 0.438(0.018) | 0.555(0.013) | 0.555(0.011) | 0.884(0.034) | 0.809(0.053) | 0.975(0.006) |
| | C = 1500 | 0.442(0.018) | 0.556(0.026) | 0.560(0.011) | 0.919(0.027) | 0.863(0.046) | 0.986(0.004) |
| | C = 2000 | 0.421(0.019) | 0.526(0.019) | 0.557(0.011) | 0.935(0.030) | 0.886(0.052) | 0.992(0.004) |

## C.2 Performance under different time window length

In this subsection, we investigate the network recovery performance under different observation window lengths, which ranges in $T \in \{1, 2, 3, 5, 10\}$.

We generate $\Theta$ and $\Psi$ following the same procedure in C.1 and fix their overlap level at about $0.30$. We generate $C = 1500$ cascade samples from transmission models Exp, Pow, Ray.

We illustrate the performance in Figure 5 and 6. In general, both transmission rate estimations and network topology recovery improve as the observation window becomes $T$ longer since effective diffusion information increases within each cascade sample. Specifically, by Figure 5, the proposed method achieves better MAE of $\Theta$ and $\Psi$ under Ray (green line with shaded one standard deviation error range) and Pow (yellow line) cascade model. MAE criterion under Exp model is not strictly monotone, but its overall trend matches that of Pow and Ray. Figure 6 shows that accuracy and precision of $\Psi$ also increases as $T$ increases and then remains stable. The results in this subsection suggest to set a large $T$ in practice, which is also consistent with our theoretical analysis.

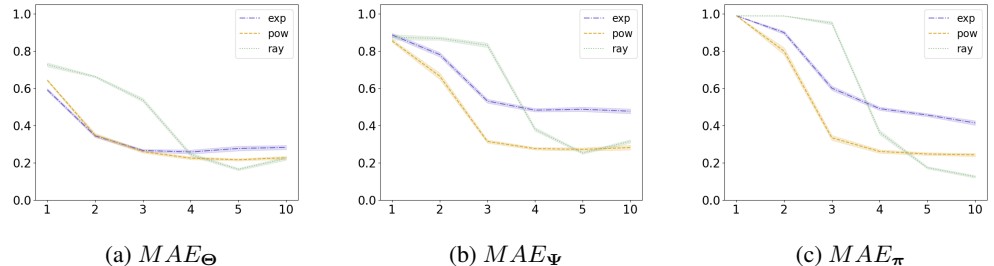

(a) $MAE_{\Theta}$      (b) $MAE_{\Psi}$      (c) $MAE_{\pi}$

Figure 5: Parameter estimation under different time window lengths.

## C.3 Network recovery without support information

In this subsection, we investigate the performance of the proposed method when support $A$ of diffusion network $\Theta$ is unobserved.

We consider diffusion networks $\Theta$ and $\Psi$ of size $N = 100$, generated by similar scheme as in C.1. Since the size of network is reduced, rank of $\Psi_1, \Psi_2$, where $\Psi = \Psi_1 \Psi_2^T$, is accordingly reduced to 15. The generated $\Theta$ and $\Psi$ have edge densities of 0.02 and 0.05, receptively. Non-zero edges of $\Theta$ and $\Psi$ have ranges 0.1 to 2 and 0.05 to 2.2, respectively. The level of overlap between the two networks is

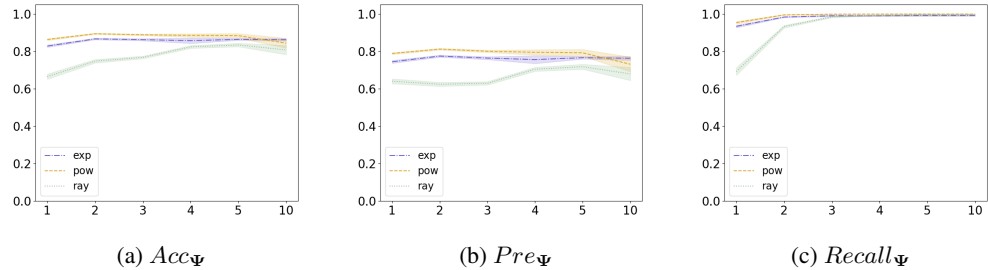

(a) $Acc_{\Psi}$          (b) $Pre_{\Psi}$          (c) $Recall_{\Psi}$

Figure 6: Topology recovery of latent network estimation under different time window lengths.

fixed at 0.15. We generate cascade samples of different sizes $C = 2000, 3000, 4000, 5000$ and from all three different transmission models Exp, Pow, and Ray.

Table 6 shows that when support of $\Theta$ is not observed, performance of the proposed method on transmission rate estimation still increases as number of samples increases for all three cascade generation models. Benchmark values in Table 6 are obtained from the proposed method with information on the support of $\Theta$. Figure 7 and 8 show the relationship between precision and recall of the estimated $\Theta$ and $\Psi$ when their entries below some varying thresholds are truncated to 0. As sample size increases, the performance of the proposed method on network topology recovery increases (shown by the larger area under P-R curves). Although the proposed method without support information of $\Theta$ cannot achieve as good performance as when the information is available, its performance on both transmission rate estimation and network topology recovery still exhibits converging pattern to the benchmark performance as sample size increases.

Table 6: Parameter estimation from the proposed method when support of $\Theta$ is unknown.

|  |  | $MAE_{\Theta}$ | $Acc_{\Theta}$ | $MAE_{\Psi}$ | $Acc_{\Psi}$ |
|---|---|---|---|---|---|
| Exp | $C = 2000$ | 0.607(0.028) | 0.628(0.013) | 0.672(0.027) | 0.803(0.015) |
|  | $C = 3000$ | 0.533(0.038) | 0.702(0.016) | 0.604(0.030) | 0.883(0.007) |
|  | $C = 4000$ | 0.447(0.107) | 0.742(0.053) | 0.520(0.089) | 0.902(0.009) |
|  | $C = 5000$ | 0.327(0.113) | 0.787(0.047) | 0.414(0.095) | 0.904(0.024) |
|  | Benchmark | 0.088(0.005) | 0.976(0.003) | 0.210(0.008) | 0.926(0.006) |
| Ray | $C = 2000$ | 0.665(0.044) | 0.573(0.021) | 0.699(0.039) | 0.601(0.020) |
|  | $C = 3000$ | 0.553(0.112) | 0.680(0.053) | 0.592(0.093) | 0.699(0.040) |
|  | $C = 4000$ | 0.292(0.054) | 0.824(0.027) | 0.376(0.042) | 0.811(0.016) |
|  | $C = 5000$ | 0.232(0.022) | 0.858(0.012) | 0.326(0.017) | 0.837(0.010) |
|  | Benchmark | 0.067(0.002) | 1.000(0.000) | 0.177(0.002) | 0.941(0.023) |
| Pow | $C = 2000$ | 0.604(0.064) | 0.732(0.026) | 0.537(0.051) | 0.855(0.010) |
|  | $C = 3000$ | 0.447(0.051) | 0.786(0.043) | 0.423(0.037) | 0.871(0.008) |
|  | $C = 4000$ | 0.331(0.062) | 0.822(0.025) | 0.344(0.048) | 0.879(0.010) |
|  | $C = 5000$ | 0.346(0.047) | 0.791(0.022) | 0.341(0.034) | 0.869(0.010) |
|  | Benchmark | 0.061(0.002) | 0.976(0.003) | 0.175(0.008) | 0.916(0.002) |

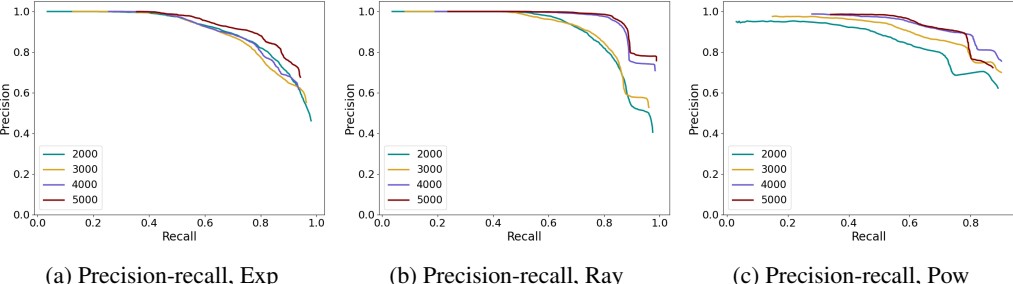

(a) Precision-recall, Exp      (b) Precision-recall, Ray      (c) Precision-recall, Pow

Figure 7: Precision-recall plot of $\Theta$

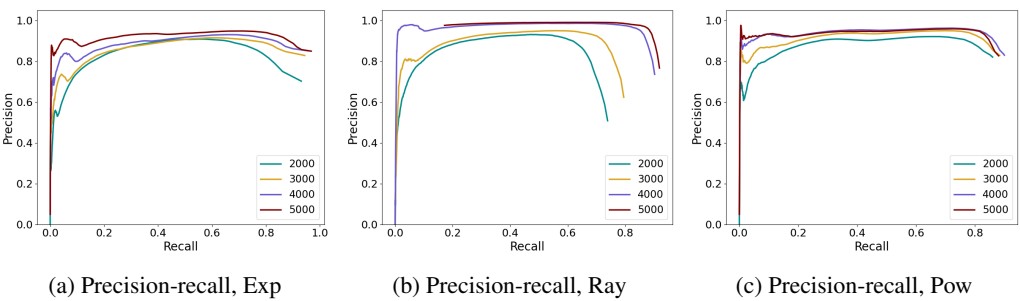

| (a) Precision-recall, Exp | (b) Precision-recall, Ray | (c) Precision-recall, Pow |

Figure 8: Precision-recall plot of $\mathbf{\Psi}$

# D    Additional details on method implementation

To give proper initializations to our proposed method, we use estimation result $\mathbf{\Lambda}$ from NetRate [31]. For experiments in Section 4.1, 4.2, C.1, and C.2, we initialize $\mathbf{\Theta_0} = \mathbf{\Lambda} \odot \mathbf{A}$ and $\mathbf{\Psi_0} = \mathbf{\Lambda} \odot (\mathbf{1} - \mathbf{A})$, where $\mathbf{A}$ is the support of $\mathbf{\Theta}$.

We employ a new initialization method in C.3 since information on $\mathbf{A}$, the support of $\mathbf{\Theta}$, is missing. We initialize $\mathbf{\Psi_0}$ as the truncated SVD of $\mathbf{\Lambda}$ at rank $r$. We then initialize $\mathbf{\Theta_0}$ as the top $\gamma\%$ entries of the residual $\mathbf{\Lambda} - \mathbf{\Psi_0}$. Both $r$ and $\gamma$ are tunable hyperparameters, where $1 \leqslant r \leqslant N$ and $0 < \gamma < 100$. In the optimization process of C.3, instead of forcing gradient updates of $\mathbf{\Theta}$ taking non-zero values only on $\mathbf{A}$, we introduce an additional $l_1$ regularizer on $\mathbf{\Theta}$ in the loss function to promote its sparsity and update $\mathbf{\Theta}$ via an additional $l_1$ soft-thresholding operator.

We also provide a short discussion in this section on the general strategy to choose the best hyperparameters $[\lambda_{\mathbf{\Theta}}, \lambda_{\mathbf{\Psi}}, \mu, \rho_{\mathbf{\Theta}}, \rho_{\mathbf{\Psi}}]$, where $\lambda$ denotes learning rate, $\mu$ denotes low-rank penalty, $\rho$ denotes $l_1$ penalty, and the subscript denotes the network associated with.

In general, as network size, sample size, or edge density increases, smaller $\lambda_{\mathbf{\Theta}}, \lambda_{\mathbf{\Psi}}$ are required for stable and good performance of the algorithm. In terms of different cascade transmission models, the optimal learning rates for Pow model have the largest magnitude, while those for Ray model have the smallest. Level of overlap, length of observation window, or the rank of $\mathbf{\Psi}$ does not significantly affect learning rate choices.

For fixed $\{\mathbf{\Theta}, \mathbf{\Psi}, \boldsymbol{\pi}\}$, there exist best threshold $\lambda_{\mathbf{\Psi}} \cdot \mu$ for singular value soft-thresholding operation and best thresholds $\lambda_{\mathbf{\Theta}} \cdot \rho_{\mathbf{\Theta}}, \lambda_{\mathbf{\Psi}} \cdot \rho_{\mathbf{\Psi}}$ for $l_1$ soft-thresholding operation. Thus, for fixed $\{\mathbf{\Theta}, \mathbf{\Psi}, \boldsymbol{\pi}\}$, we need to adjust penalties inversely proportional to learning rates for the best performance of the proposed algorithm. $l_1$ penalties should be increased as transmission rates on $\mathbf{\Theta}, \mathbf{\Psi}$ increase, while $\mu$ should be increased as rank of $\mathbf{\Psi}$ increases.

# E    Proofs for theoretical results

### Proposition 1

*Proof.* To prove the column-wise identification between $\mathbf{\Theta}_{\cdot i}$ and $\mathbf{\Psi}_{\cdot i}$, we only need to consider node $i$'s direct parents in two networks as $\mathcal{R}_i^{\mathbf{\Theta}} := \{j \in \{1, \cdots, N\} \mid \mathbf{\Theta}_{ji} > 0\}$ and $\mathcal{R}_i^{\mathbf{\Psi}} := \{j \in \{1, \cdots, N\} \mid \mathbf{\Psi}_{ji} > 0\}$ as the sets of nodes that can directly reach node $i$ on $\mathbf{\Theta}$ and $\mathbf{\Psi}$. Denote the activation times of the direct parents of node $i$ as $\boldsymbol{t}_{\mathcal{R}_i} \in [0, T]^{|\mathcal{R}_i^{\mathbf{\Theta}} \cup \mathcal{R}_i^{\mathbf{\Psi}}|} := \{t_j \mid j \in \mathcal{R}_i^{\mathbf{\Theta}} \cup \mathcal{R}_i^{\mathbf{\Psi}}\}$.

If $\pi_i \boldsymbol{P}_I(t; \mathbf{\Theta}_{\cdot i}) + (1 - \pi_i)\boldsymbol{P}_I(t; \mathbf{\Psi}_{\cdot i}) = \tilde{\pi}_i \boldsymbol{P}_I(t; \tilde{\mathbf{\Theta}}_{\cdot i}) + (1 - \tilde{\pi}_i)\boldsymbol{P}_I(t; \tilde{\mathbf{\Psi}}_{\cdot i})$, then we have

$$\pi_i \sum_j H(t_j; \mathbf{\Theta}_{ji}) \prod_j S(t_j; \mathbf{\Theta}_{ji}) + (1 - \pi_i) \sum_j H(t_j; \mathbf{\Psi}_{ji}) \prod_j S(t_j; \mathbf{\Psi}_{ji}) \quad (15)$$

$$= (1 - \tilde{\pi}_i) \sum_j H(t_j; \tilde{\mathbf{\Theta}}_{ji}) \prod_j S(t_j; \tilde{\mathbf{\Theta}}_{ji}) + (1 - \tilde{\pi}_i) \sum_j H(t_j; \tilde{\mathbf{\Psi}}_{ji}) \prod_j S(t_j; \mathbf{\Psi}_{ji}), \quad (16)$$

where the summation and multiplication are over $j \in \mathcal{R}_i^{\mathbf{\Theta}} \cup \mathcal{R}_i^{\mathbf{\Psi}}$ with $p := |\mathcal{R}_i^{\mathbf{\Theta}} \cup \mathcal{R}_i^{\mathbf{\Psi}}|$. Given that survival function satisfies $S(t_j; \lambda_{ji}) = \exp\{\lambda_{ji} h(t_i - t_j)\}$ for some non-negative function $h(\cdot)$, then

the infection likelihood and hazard function are

$$f(t_j; \lambda_{ji}) = \lambda_{ji} \exp\{-\lambda_{ji} h(t_i - t_j)\} h'(t_i - t_j); \; H(t_j; \lambda_{ji}) = \lambda_{ji} h'(t_i - t_j).$$

Denote $w_1(t_j) = \pi_i h'(t_i - t_j) \prod_k S(t_k; \boldsymbol{\Theta}_{ki})$, $w_2(t_j) = (1 - \pi_i) h'(t_i - t_j) \prod_k S(t_k; \boldsymbol{\Psi}_{ki})$, $\tilde{w}_1(t_j) = \tilde{\pi}_i h'(t_i - t_j) \prod_k S(t_k; \tilde{\boldsymbol{\Theta}}_{ki})$, and $\tilde{w}_2(t_j) = (1 - \tilde{\pi}_i) h'(t_i - t_j) \prod_k S(t_k; \tilde{\boldsymbol{\Psi}}_{ki})$. Then (15) can be re-written as

$$\sum_j w_1(t_j) \boldsymbol{\Theta}_{ji} + \sum_j w_2(t_j) \boldsymbol{\Psi}_{ji} - \sum_j \tilde{w}_1(t_j) \tilde{\boldsymbol{\Theta}}_{ji} - \sum_j \tilde{w}_2(t_j) \tilde{\boldsymbol{\Psi}}_{ji} = 0, \tag{17}$$

which should hold for any $\boldsymbol{t}_{\mathcal{R}_i}$. Consider a $4p \times 4p$ matrix $\boldsymbol{\Gamma}$ with each row being $[w_1(t_1), \cdots, w_1(t_p), w_2(t_1), \cdots, w_2(t_p), \tilde{w}_1(t_1), \cdots, \tilde{w}_1(t_p), \tilde{w}_2(t_1), \cdots, \tilde{w}_2(t_p)]$ where each row takes different values of $\boldsymbol{t}_{\mathcal{R}_i}$. Based on assumption that there exist $j^* \neq i$ such that $\boldsymbol{\Theta}_{j^*i} \neq \boldsymbol{\Psi}_{j^*i}$, then up to permutation between $\tilde{\boldsymbol{\Theta}}_{\cdot i}$ and $\tilde{\boldsymbol{\Psi}}_{\cdot i}$, the columns corresponding to different $t_j$ will be linear independent. Without loss of generality, we only need to investigate the $4p \times 4$ submatrix $\boldsymbol{\Gamma}_j$ with each row being $[w_1(t_j), w_2(t_j), \tilde{w}_1(t_j), \tilde{w}_2(t_j)]$. We prove that $w_1(t_j) = \tilde{w}_1(t_j), w_2(t_j) = \tilde{w}_2(t_j)$ and $\boldsymbol{\Theta}_{ji} = \tilde{\boldsymbol{\Theta}}_{ji}$, $\boldsymbol{\Psi}_{ji} = \tilde{\boldsymbol{\Psi}}_{ji}$ by contradiction argument. For two different values $t_j$ and $t'_j$, we have

$$\frac{w_1(t_j)}{w_1(t'_j)} = \frac{w_2(t_j)}{w_2(t'_j)} \Rightarrow \frac{\prod_j S(t_j, \boldsymbol{\Theta}_{ji})}{\prod_j S(t'_j, \boldsymbol{\Theta}_{ji})} = \frac{\prod_j S(t_j, \boldsymbol{\Psi}_{ji})}{\prod_j S(t'_j, \boldsymbol{\Psi}_{ji})} \tag{18}$$

$$\Rightarrow \sum_j (\boldsymbol{\Theta}_{ji} - \boldsymbol{\Psi}_{ji})(h(t_i - t_j) - h(t_i - t'_j)) = 0, \tag{19}$$

and we can choose $t_j \neq t'_j$ for those $j^*$ such that $\boldsymbol{\Theta}_{j^*i} \neq \boldsymbol{\Psi}_{j^*i}$. Then the above equation implies $\boldsymbol{\Theta}_{j^*i} = \boldsymbol{\Psi}_{j^*i}$, which cause contradiction. Therefore, the rank of $\boldsymbol{\Gamma}_j$ is larger than 1.

If there exists $a, b$ such that $a \times w_1(t_j) + b \times w_2(t_j) = \tilde{w}_1(t_j)$ hold for any $\boldsymbol{t}_{\mathcal{R}_i}$. Then we let $t_j = t_i$ for all $j \in \mathcal{R}_i^{\boldsymbol{\Theta}} \cup \mathcal{R}_i^{\boldsymbol{\Psi}}$ except $j^*$. Then we have

$$a \times \pi_i h'(t_i - t_{j*}) \exp\{h(t_i - t_{j*}) \boldsymbol{\Theta}_{j*i}\} + b \times (1 - \pi_i) h'(t_i - t_{j*}) \exp\{h(t_i - t_{j*}) \boldsymbol{\Psi}_{j*i}\} \tag{20}$$

$$= \tilde{\pi}_i h'(t_i - t_{j*}) \exp\{h(t_i - t_{j*}) \tilde{\boldsymbol{\Theta}}_{j*i}\} \tag{21}$$

$$\Rightarrow a' \exp\{h(t_i - t_{j*})(\boldsymbol{\Theta}_{j*i} - \tilde{\boldsymbol{\Theta}}_{j*i})\} + b' \exp\{h(t_i - t_{j*})(\boldsymbol{\Psi}_{j*i} - \tilde{\boldsymbol{\Theta}}_{j*i})\} = 1, \tag{22}$$

where $a' = a \frac{\pi_i}{\tilde{\pi}_i}, b' = b \frac{1 - \pi_i}{\tilde{\pi}_i}$. Notice that at least one of $\boldsymbol{\Theta}_{j*i} - \tilde{\boldsymbol{\Theta}}_{j*i}$ and $\boldsymbol{\Psi}_{j*i} - \tilde{\boldsymbol{\Theta}}_{j*i}$ is not zero and the monotonicity of $\exp(\lambda t)$ in terms of $t$, the above exponential function equation cannot hold as long as $h(t_i - t)$ is not constant over $t$. Therefore, the rank of $\boldsymbol{\Gamma}_j$ can not be 2. Using the same argument above, we can show the rank of $\boldsymbol{\Gamma}_j$ can not be 3 as well.

If the rank of $\boldsymbol{\Gamma}_j$ is 4, i.e., $\boldsymbol{\Gamma}_j$ is full rank, then $\boldsymbol{\Gamma}$ is also full rank by applying above argument on each $j \in \mathcal{R}_i^{\boldsymbol{\Theta}} \cup \mathcal{R}_i^{\boldsymbol{\Psi}}$ except $j^*$. Based on (17), we have $\boldsymbol{\Gamma}[\boldsymbol{\Theta}_{\cdot i}^\top, \boldsymbol{\Psi}_{\cdot i}^\top, \tilde{\boldsymbol{\Theta}}_{\cdot i}^\top, \tilde{\boldsymbol{\Psi}}_{\cdot i}^\top]^\top = \boldsymbol{0}$, which leads to $\boldsymbol{\Theta}_{\cdot i}^\top = \boldsymbol{\Psi}_{\cdot i}^\top = \tilde{\boldsymbol{\Theta}}_{\cdot i}^\top = \tilde{\boldsymbol{\Psi}}_{\cdot i}^\top = \boldsymbol{0}$ and contradicts to assumption $\|\boldsymbol{\Theta}_{\cdot i}\|_1 + \|\boldsymbol{\Psi}_{\cdot i}\|_1 > 0$. Therefore, $w_1(t_j) = \tilde{w}_1(t_j), w_2(t_j) = \tilde{w}_2(t_j)$ and $\boldsymbol{\Theta}_{ji} = \tilde{\boldsymbol{\Theta}}_{ji}$, $\boldsymbol{\Psi}_{ji} = \tilde{\boldsymbol{\Psi}}_{ji}$ for all $j$. Finally, by the definition of $w_1(t_j)$ and $tildew_1(t_j)$, we can derive $\pi_i = \tilde{\pi}_i$. The statement based on $\boldsymbol{P}_U$ can be similarly derived. $\square$

**Proposition 2**

*Proof.* Given Proposition 1, we have the column-wise identification of $\boldsymbol{\Theta}_{\cdot i}$ and $\boldsymbol{\Psi}_{\cdot i}$, and only the identification of $\boldsymbol{\Theta}$ and $\boldsymbol{\Psi}$ up to the column exchange between $\boldsymbol{\Theta}$ and $\boldsymbol{\Psi}$. Notice that Proposition 1 guarantee the identification of $\boldsymbol{C} = \boldsymbol{\Theta} + \boldsymbol{\Psi}$. Then if

$$\max_{N \in \boldsymbol{\Lambda_1}(\boldsymbol{\Theta}), \|N\|_\infty \leqslant 1} \|N\|_2 \times \max_{N \in \boldsymbol{\Lambda_2}(\boldsymbol{\Psi}), \|N\|_2 \leqslant 1} \|N\|_\infty < 1, \tag{23}$$

we have $\boldsymbol{\Lambda_1}(\boldsymbol{\Theta}) \cap \boldsymbol{\Lambda_2}(\boldsymbol{\Psi}) = \varnothing$ based on Proposition 1 in [6]. Notice that $\boldsymbol{\Theta} \in \boldsymbol{\Lambda_1}(\boldsymbol{\Theta})$ and $\boldsymbol{\Psi} \in \boldsymbol{\Lambda_2}(\boldsymbol{\Psi})$, then $\boldsymbol{\Theta}$ and $\boldsymbol{\Psi}$ are identifiable conditioning on $\boldsymbol{C}$ is identifiable. $\square$

**Theorem 3.1**

*Proof.* We first show $\boldsymbol{\Xi_\Theta}(s)$, $\boldsymbol{\Xi_\Psi}(\rho)$, and $\boldsymbol{\Xi_\pi}$ are convex sets for any $s > 0$, $\rho > 0$ by checking that the convex combination of any two elements in $\boldsymbol{\Xi_\Theta}(s)$, $\boldsymbol{\Xi_\Psi}(\rho)$, or $\boldsymbol{\Xi_\pi}$ still lies in the set.

For any $\boldsymbol{\Theta_1}, \boldsymbol{\Theta_2} \in \boldsymbol{\Xi_\Theta}(s)$ and $0 \leqslant \lambda \leqslant 1$,

$$0 \leqslant \lambda(\boldsymbol{\Theta_1})_{ij} + (1-\lambda)(\boldsymbol{\Theta_2})_{ij} \leqslant \lambda\beta_1 + (1-\lambda)\beta_1 = \beta_1 \tag{24}$$

$$(\lambda\boldsymbol{\Theta_1} + (1-\lambda)\boldsymbol{\Theta_2}) \odot (\boldsymbol{I} - \boldsymbol{A}) = \lambda[\boldsymbol{\Theta_1} \odot (\boldsymbol{I} - \boldsymbol{A})] + (1-\lambda)[\boldsymbol{\Theta_2} \odot (\boldsymbol{I} - \boldsymbol{A})] = \boldsymbol{0}$$

$\boldsymbol{\Xi_\Theta}(s)$ is thus a convex set. Note that $s > 0$ avoids the trivial case where $\boldsymbol{A}$ is a zero matrix.

It is easy to argue that $\boldsymbol{\Xi_\pi}$ is a convex set using similar argument to (24).

For any $\boldsymbol{\Psi_1}, \boldsymbol{\Psi_2} \in \boldsymbol{\Xi_\Psi}(\rho)$ and $0 \leqslant \lambda \leqslant 1$,

$$||\lambda\boldsymbol{\Psi_1} + (1-\lambda)\boldsymbol{\Psi_2}||_* \leqslant \lambda||\boldsymbol{\Psi_1}||_* + (1-\lambda)||\boldsymbol{\Psi_2}||_* = \rho \tag{25}$$

By (25) and similar argument to (24), $\boldsymbol{\Xi_\Psi}(\rho)$ is a convex set.

Next, we show $\boldsymbol{Q}_3(\boldsymbol{\pi} \mid \boldsymbol{\Omega}^{(s)})$ is concave over $\boldsymbol{\pi}$. Recall the formulation of $\boldsymbol{Q}_3(\boldsymbol{\pi} \mid \boldsymbol{\Omega}^{(s)})$ in A.1

$$\boldsymbol{Q}_3(\boldsymbol{\pi} \mid \boldsymbol{\Omega}^{(s)}) = \frac{1}{C} \sum_{c=1}^{C} [(\hat{\boldsymbol{\pi}}^c)^T \log(\boldsymbol{\pi}) + (\boldsymbol{1} - \hat{\boldsymbol{\pi}}^c)^T \log(\boldsymbol{1} - \boldsymbol{\pi})]$$

When $\boldsymbol{\pi} \in \boldsymbol{\Xi_\pi}$, both $\log(\boldsymbol{\pi})$ and $\log(\boldsymbol{1} - \boldsymbol{\pi})$ are concave over $\boldsymbol{\pi}$. From the above formulation, $\boldsymbol{Q}_3(\boldsymbol{\pi} \mid \boldsymbol{\Omega}^{(s)})$ is concave over $\boldsymbol{\pi}$ by linearity of concavity.

Without loss of generality, we show $\boldsymbol{Q}_1(\boldsymbol{\Theta} \mid \boldsymbol{\Omega}^{(s)})$ is concave on $\boldsymbol{\Theta}$ under the assumption that hazard function $H(t \mid t', \lambda)$ satisfies $\partial H^2(t \mid t', \lambda)/\partial\lambda^2 = 0$ for $t \geqslant t'$.

Note that this assumption immediately implies the concavity of $H(t \mid t', \lambda)$ and the log concavity of $S(t \mid t', \lambda)$ over $\lambda$.

Recall the definition of $\boldsymbol{P}_I$ and $\boldsymbol{P}_U$ in Section 3.1 we have

$$\log \boldsymbol{P}_I(t_i; \boldsymbol{\Theta}_{\cdot i}) = \log\Big( \sum_{j:t_j<t_i} H(t_i \mid t_j, \boldsymbol{\Theta}_{ji}) \prod_{k:t_k<t_i} S(t_i \mid t_k, \boldsymbol{\Theta}_{ki})\Big) \tag{26}$$

$$\log \boldsymbol{P}_U(T; \boldsymbol{\Theta}_{\cdot i}) = \log\Big( \prod_{j:t_j<T} S(T \mid t_j, \boldsymbol{\Theta}_{ji})\Big) \tag{27}$$

Rewrite (26) and (27) in the following form,

$$\log \boldsymbol{P}_I(t_i; \boldsymbol{\Theta}_{\cdot i}) = \log\Big(\sum_j H(t_i \mid t_j, \boldsymbol{\Theta}_{ji})\Big) + \sum_k \log S(t_i \mid t_k, \boldsymbol{\Theta}_{ki})$$

$$\log \boldsymbol{P}_U(T; \boldsymbol{\Theta}_{\cdot i}) = \sum_j \log S(T \mid t_j, \boldsymbol{\Theta}_{ji})$$

Then by concavity and monotonicity of log function, composition rule for concavity, and linearity of concavity, both $\log \boldsymbol{P}_I(t_i; \boldsymbol{\Theta}_{\cdot i})$ and $\log \boldsymbol{P}_U(T; \boldsymbol{\Theta}_{\cdot i})$ are concave over $\boldsymbol{\Theta}_{\cdot i}$.

Concavity of $\boldsymbol{Q}_1(\boldsymbol{\Theta} \mid \boldsymbol{\Omega}^{(s)})$ over $\boldsymbol{\Theta}$ immediately follows from composition rule for concavity and the proof for $\boldsymbol{Q}_3(\boldsymbol{\pi} \mid \boldsymbol{\Omega}^{(s)})$. Concavity of $\boldsymbol{Q}_2(\boldsymbol{\Theta} \mid \boldsymbol{\Psi}^{(s)})$ directly follows this proof by changing $\boldsymbol{\Theta}$ to $\boldsymbol{\Psi}$.

By the above proof, the linear combination $\sum_{j:t_j^c>T} \hat{\pi}_j^c \log \boldsymbol{P}_U(T; \boldsymbol{\Theta}_{\cdot j})$ where $\hat{\pi}_j^c$ depends on $\boldsymbol{\Omega}^{(s)}$ instead of $\boldsymbol{\Theta}$ is concave over $\boldsymbol{\Theta}$. Therefore, $\mathbb{E}_{\boldsymbol{t}}(\sum_{j:t_j^c>T} \hat{\pi}_j^c \log \boldsymbol{P}_U(T; \boldsymbol{\Theta}_{\cdot j}))$ is concave in terms of $\boldsymbol{\Theta}$. In addition, notice that $\frac{\partial^2 \mathbb{E}_{\boldsymbol{t}}[\boldsymbol{Q}_1(\boldsymbol{\Theta}|\boldsymbol{\Omega}^{(s)})]}{\partial^2\boldsymbol{\Theta}}$ is a $(N^2 - N) \times (N^2 - N)$ diagonal block matrix as $\frac{\partial^2 \mathbb{E}_{\boldsymbol{t}}[\boldsymbol{Q}_1(\boldsymbol{\Theta}|\boldsymbol{\Omega}^{(s)})]}{\partial\boldsymbol{\Theta}_{ij}\partial\boldsymbol{\Theta}_{kl}} = 0$ if $j \neq l$, which leads to $N$ blocks $\Big(\frac{\partial^2 \mathbb{E}_{\boldsymbol{t}}[\boldsymbol{Q}_1(\boldsymbol{\Theta}|\boldsymbol{\Omega}^{(s)})]}{\partial\boldsymbol{\Theta}_{ji}\partial\boldsymbol{\Theta}_{ki}}\Big)_{(N-1)\times(N-1)}$, $i = 1, \cdots, N$. And the $i$th block corresponds to the second derivative of

$$\frac{1}{C} \sum_{c=1}^{C} \hat{\pi}_i^c \Big(\boldsymbol{1}_{\{t_i^c \leqslant T\}} \log \boldsymbol{P}_I(t_i; \boldsymbol{\Theta}_{\cdot i}) + \boldsymbol{1}_{\{t_i^c > T\}} \log \boldsymbol{P}_U(T; \boldsymbol{\Theta}_{\cdot i})\Big)$$

Based on the above arguments, we only need to show that the strictly concavity for each block in $\frac{\partial^2 \mathbb{E}_t[\boldsymbol{Q}_1(\boldsymbol{\Theta}|\boldsymbol{\Omega}^{(s)})]}{\partial^2 \boldsymbol{\Theta}}$, i.e., $\mathbb{E}_t\big(\frac{1}{C}\sum_{c=1}^{C}\mathbf{1}_{\{t_i^c \leqslant T\}}\hat{\pi}_i^c \log \boldsymbol{P}_I(t_i;\boldsymbol{\Theta}_{\cdot i})\big)$. We then set $C$ to be larger enough such that $C_0 := \sum_{c=1}^{C}\mathbf{1}_{\{t_i^c \leqslant T\}} > N-1$ and set the index of cascade where $i$ is infected as $1, \cdots, C_0$, and

$$\mathbb{E}_t\big(\frac{1}{C}\sum_{c=1}^{C}\mathbf{1}_{\{t_i^c \leqslant T\}}\hat{\pi}_i^c \log \boldsymbol{P}_I(t_i;\boldsymbol{\Theta}_{\cdot i})\big) = \mathbb{E}_t\big(\frac{1}{C}\sum_{c=1}^{C_0}\hat{\pi}_i^c \log \boldsymbol{P}_I(t_i;\boldsymbol{\Theta}_{\cdot i})\big)$$

Then we denote $\mathcal{Q} = \frac{1}{C}\sum_{c=1}^{C_0}\hat{\pi}_i^c \log \boldsymbol{P}_I(t_i;\boldsymbol{\Theta}_{\cdot i})$ and follow the argument of Lemma 10 in [8] to prove $\mathcal{Q}$ is positive definite. Following their notations and $\alpha := \boldsymbol{\Theta}_{\cdot i}$, we can write

$$\mathcal{Q} = \boldsymbol{D}(\alpha) + \frac{1}{C}\boldsymbol{X}(\alpha)[\boldsymbol{X}(\alpha)]^\top$$

where $\boldsymbol{D}(\alpha) = \frac{1}{C}\sum_{c=1}^{C_0}\boldsymbol{D}(t^c;\alpha)$ and $\boldsymbol{D}(t^c;\alpha)$ is a diagonal matrix with $[\boldsymbol{D}(t^c;\alpha)]_{jj} = -\hat{\pi}_i^c S''(t_i^c \mid t_j^c;\alpha_k) - \hat{\pi}_i^c h^{-1}(t^c,\alpha)H''(t_i^c \mid t_j^c;\alpha_k)$, and $h(t^c,\alpha) = \sum_{j:t_j^c < t_i^c} H(t_i^c \mid t_j^c;\alpha_k)$. In addition, $\boldsymbol{X}(\alpha)$ is a $(N-1)$-by-$C_0$ matrix as

$$\boldsymbol{X}(\alpha) = [\boldsymbol{X}(t^1;\alpha) \mid \boldsymbol{X}(t^1;\alpha) \mid \cdots, \boldsymbol{X}(t^{C_0};\alpha)],$$

where each column $\boldsymbol{X}(t^c;\alpha) = \sqrt{\hat{\pi}_i^c}h^{-1}(t^c,\alpha)\nabla_\alpha h(t^c,\alpha)$. Given the assumption that the probabilities of being source node $\boldsymbol{P}(v) > 0$ for $v \in \mathcal{R}$ where $\mathcal{R}$ denotes the set of nodes from which $i$ is reachable via a directed path, we can follow the same argument in Lemma 10 of [8] to show $\mathcal{Q}$ is strictly concave in terms of $\boldsymbol{\Theta}$. Then $\mathbb{E}_t(\mathcal{Q})$ is also strictly concave. The strong concavity of $\mathbb{E}_t[\boldsymbol{Q}_2(\boldsymbol{\Psi} \mid \boldsymbol{\Omega}^{(s)})]$ in terms of $\boldsymbol{\Psi}$ can be similarly proved. $\qquad\square$

# F   Information on computer resources

Numerical experiments in this paper were carried out in Google Colab on a single Nvidia A100 GPU with 80GB of memory available. Regardless of failed experiments, the entire numerical experiment section took approximately 50 hours of computation time.

In addition, to evaluate the benchmark algorithm ConNIe, we used SNOPT (version 7.7) as the underlying constrained optimization solver [15, 16].

