# OpenReview forum: "Diffusion Network Inference for Cross-layer Cascades"
_NeurIPS.cc/2025/Conference — NeurIPS 2025 poster_

### Official Review · Reviewer_LD4Y · 2025-07-02

**Clarity:** 3
**Significance:** 3
**Originality:** 3
**Rating:** 4
**Confidence:** 3

**Summary:**

This paper presents a novel double network mixture model to infer latent diffusion structures from observed cascade data, in settings where cascading behavior is governed by heterogeneous and possibly unobserved pathways. The key idea is to model the diffusion process as a mixture over two networks—one possibly observed and one latent—by introducing node-wise, cascade-specific Bernoulli indicators. The authors propose an EM-based estimation framework with convex relaxation techniques (nuclear norm), derive identifiability conditions, and provide both statistical guarantees and empirical validation through simulations and real-world sociology publication data.

**Questions:**

1. **Posterior variance of $\hat{\pi}_i^c$:** The estimation of $\hat{\pi}_i^c$ is treated as a point estimate (Eq. 10). Can the authors elaborate on the uncertainty of these estimates, or propose a way to obtain variance/confidence intervals?

2. **Scalability:** The authors acknowledge EM’s scalability limitations in the discussion. It would help if they can give a rough estimate of runtime or memory complexity as N grows. Also: is it possible to parallelize E-step across cascades?

3. **Interpretation of π vector:** In real data, what does a high π\_i mean in practice? Would be nice to provide a short discussion of how π correlates with observable university features (e.g., ranking, location).

4. **ρ selection:** How is ρ (the nuclear norm constraint) chosen in practice? Is there a validation process, or is it fixed?

5. **Real-world validation:** In the sociology case study, results are interesting but more qualitative validation (e.g., overlap with known collaborations or conference co-membership) would make the latent network interpretation more convincing.

**Ethical Concerns:**

["NO or VERY MINOR ethics concerns only"]

**Final Justification:**

I have read the rebuttal and seen their new experiments that have convinced me their work is ready for publication.

**Limitations:**

Yes

**Paper Formatting Concerns:**

None major. There are a few grammar slips and minor typesetting inconsistencies (e.g., spacing around symbols, repeated words like “of of” in Section 5), but not a big deal.

**Quality:**

3

**Strengths And Weaknesses:**

Strengths: The paper tackles a relevant and underexplored problem: inferring heterogeneous diffusion structures when cascades spread through multiple unobserved mechanisms. The methodology is sound, well-formulated mathematically, and supported by solid theoretical guarantees (identifiability, convexity of optimization). The empirical evaluation is quite extensive, including different network topologies and transmission models. The clarity is generally clear, although not always very fluid to read. A few symbols (e.g., $\Lambda$) are overloaded, and figure captions could be more descriptive. The contribution is significant to the field of network inference and cascade modeling.  Application to sociological topic diffusion also illustrates a valuable real-world use case.

No major weaknesses to me at this point. See questions below.

---

> ### Author Rebuttal · Authors · 2025-07-31
>
> We really appreciate the reviewer’s thoughtful feedback, which has significantly improved our paper! In the following response, we discuss and justify the implications of our assumptions, and demonstrate our method's generalizability with extensive experiments.
>
> **Q1**
>
> Unlike model parameters $\bf{\Theta}$ and $\bf{\Psi}$ and $\boldsymbol{\pi}$, $\hat{\pi}_i^c$ indicates the propability of unit $i$ choosing diffusion netork $\boldsymbol{\Theta}$ at the $c$-th cascade, which is treated as part of the missing data in EM algorithm rather than parameters we truly wish to estimate. In other words, we do not have replicate sample to estimate the variance $\hat{\pi}_i^c$ since it changes over both unit sample $i$ and cascade sample $c$. On the other hand, we can estimate the uncertainty of $\hat{\pi}_i$, which is provided in the tables as standard deviations.
>
> **Q2**
>
> We agree that EM algorithm is not very scalable in general. However, although our optimization is formulated as EM-type update, we argue that the implicit convex nature of our objective function allows the optimization to be as efficient as gradient descent, which is fundamentally different to classic EM, and therefore scalable to large network setting.
>
> The first reason for the poor scalability of classic EM algorithm is the posterior inference in the E-step, which may suffer from low computational efficiency due to sampling and numerical integration. However, as we pointed out in section 3.4, the E-step in our algorithm has an explicit form in equation (10), and therefore can be efficiently calculated without using parallel computing.
>
> The second scalability issue of classic EM is that the optimization in M-step may be expansive. However, we have proved in Theorem 3.1 that the objective function $\boldsymbol{Q}(\boldsymbol{\Omega})$ in the M-step is strictly concave, which makes the optimization as efficient as gradient descent. In addition, in each EM-iteration we only perform gradient ascent one time instead of maximizing $\boldsymbol{Q}(\boldsymbol{\Omega})$ until convergence, where $\boldsymbol{\Omega} = \\{\boldsymbol{\Theta},\boldsymbol{\Psi},\boldsymbol{\pi}\\}$ are our model parameters.
> It is been proved in [1] that the above EM-update is equivalent to gradient decent given strictly concave and well-behaved $\boldsymbol{Q}(\boldsymbol{\Omega})$, and therefore it can be proved that our optimization enjoys the geometric convergence rate as
>
> $$
> \\|  \boldsymbol{\Omega}^{(s)} - \boldsymbol{\Omega}^{\star} \\| \leq \rho^s \\|  \boldsymbol{\Omega}^{(0)} - \boldsymbol{\Omega}^{\star}\\|,
> $$
>
> for some $\rho<1$, where $\boldsymbol{\Omega}^{(s)} $ are the estimation from the $s$-th update, and $\boldsymbol{\Omega}^{\star}$ are the true parameter values. The rate equivalently fast to that of gradient descent.
>
> Last but not least, when handling large networks, we can replace the standard SVD operation in the $M$-step by randomized SVD or Lanczos algorithm. Then, we can further reduce the computational cost from $\mathcal{O}(N^3)$ to $\mathcal{O}(rN^2)$ where $r<<N$ is the rank of network $\boldsymbol{\Psi}$ and $N$ is the size of network (or in practice, $r$ + $k$ where $k$ is usually 10 to 20 for more stable convergence). This is at the same order of matrix multiplication between two low-rank matrix. \textbf{The overall computational complexity of our algorithm is: $T \cdot O(N^2 \cdot M) + T \cdot O(N^2 \cdot r)$}, where $T$ is the number of iterations executed and $M$ is sample size.
>
> Besides the above justification of our method's scalability from a theoretic perspective, we also perform numerical experiments to proof it. To handle possible memory constraints when network size is large, we use a mini-batch version of our algorithm such that each update of both E-step and M-step are conducted using $B$ samples where $B$ is batch size. In on complete iteration, we make $\lceil M/B \rceil$ updates so that all $M$ samples are fully used.
>
> In our first experiment, we fix total number of iterations $T = 100$, sample size $M = 5000$, batch size $B = 40$, dimension $r + k = 20$ for randomized SVD, and only change $N = 500, 1000, 2000, 4000$. We report the mean and the standard deviation of execution time in second per iteration for all four different network sizes in the following table.
>
> | Size          | 500         | 1000        | 2000        | 4000        |
> |:-------------:|:-----------:|:-----------:|:-----------:|:-----------:|
> | Time (s/iter) | 0.032(0.002)| 0.952(0.004)| 3.685(0.001)| 14.574(0.006)|
>
> **Table 1** Execution Time Per Iteration under Different Node Sizes (mean ± std).
>
> We see that when network size $N$ is large enough for GPU kernels to start fully synchronize, the execution time starts to agree with the theoretical computation complexity of our algorithm: As $N$ doubles, execution time per iteration increases by a factor of four. Note that if more memory is available, we can increase the batch size $B$ accordingly for better parallelization on GPU and thus shorter execution time per iteration. Please see response to reviewer 3 for more numerical experiments related to the scalability of our method.
>
> **Q3**
>
> If we code $Z=1$ to mean diffusion via network $\boldsymbol{\Theta}$, and $Z=0$ for network $\boldsymbol{\Psi}$, then high $\pi_i$ indicates the $i$-th unit more prefers to interact with cascade via network $\boldsymbol{\Theta}$ and vice versa. In our application, a high $\pi_i$ indicates that the institution $i$ have higher probability to communicate research topic via latent academic network instead of geographic network. We have studied the relation between $\pi_i$ with university rankings and found that high-ranked universities have higher tendency of exchanging idea via latent network. The average $\pi_i$ among the 104 universities is $0.601$, indicating that research topics diffuse via latent network more frequently than via geographic network. Due to the space limit, we will add the additional analysis in Supplementary.
>
> **Q4**
>
> We choose the nuclear norm constraint based on cross-validation in practice. Here is also a short discussion on the general strategy to choose the best hyperparameters
>
> $[\lambda_{\boldsymbol{\Theta}}, \lambda_{\boldsymbol{\Psi}}, \mu, \rho_{\boldsymbol{\Theta}}, \rho_{\boldsymbol{\Psi}}]$,
>
> where $\lambda$ denotes learning rate, $\mu$ denotes low-rank penalty, $\rho$ denotes $l_1$ penalty, and the subscript denotes the network associated with. In general, as network size, sample size, or edge density increases, smaller
>
> $\lambda_{\boldsymbol{\Theta}}, \lambda_{\boldsymbol{\Psi}}$
>
> are required for stable and good performance of the algorithm.
>
> For fixed $\{\boldsymbol{\Theta}, \boldsymbol{\Psi}, \boldsymbol{\pi}\}$,  there exist best threshold $\lambda_{\boldsymbol{\Psi}} \cdot\mu$ for singular value soft-thresholding operation and best thresholds
>
> $\lambda_{\boldsymbol{\Theta}} \cdot \rho_{\boldsymbol{\Theta}}$, $\lambda_{\boldsymbol{\Psi}} \cdot \rho_{\boldsymbol{\Psi}}$
>
> for $l_1$ soft-thresholding operation. Thus, for fixed $\{\boldsymbol{\Theta}, \boldsymbol{\Psi}, \boldsymbol{\pi}\}$, we need to adjust penalties inversely proportional to learning rates for the best performance of the proposed algorithm. $l_1$ penalties should be increased as transmission rates on $\boldsymbol{\Theta}, \boldsymbol{\Psi}$ increase, while $\mu$ should be increased as rank of $\boldsymbol{\Psi}$ increases.
>
> **Q5**
>
> We agree with the reviewer on this point in general. Notice that the dataset involves information and authors from $29,725$ unique articles from 23 top generalist sociology journal. Extracting all the authors and preprocessing their collaborations relations and conference co-membership can be intensively time-consuming. Therefore, we focus more on introducing and illustrating a new cascade analysis method in this paper, and will add more qualitative validation as future works.
>
> References:
>
> [1] Statistical guarantees for the EM algorithm: From population to sample-based analysis

---

> > ### Author Response · Authors · 2025-08-05
> >
> > Dear Reviewer LD4Y,
> >
> > We would like to offer an additional clarification and apologize for any potential confusion: our reference to “Reviewer 3” should have been “Reviewer q7jT.” If you are interested, the additional scalability experiments can be found in Q1 and Q2 of Reviewer q7jT’s response. Thank you for your understanding and we apologize again for any inconvenience.

---

> > > ### Author Response · Authors · 2025-08-08
> > >
> > > Dear Reviewer LD4Y,
> > >
> > > We have provided these additional comments—building on our earlier rebuttal—in hopes of facilitating further discussion. We would be grateful for any insights or questions you might have.
> > >
> > > **Additional Notes on Q1**
> > >
> > > Despite an exhaustive review of the literature, we found no established methods for directly quantifying uncertainty in $\hat{\pi}_i^c$. We further clarify why a confidence interval for $\hat{\pi}_i^c$ is typically neither targeted nor well defined. Note that
> > >
> > > \begin{align*}
> > >     \hat{\pi}^c_ i := \boldsymbol{P}(Z_ i^c = 1 \mid \boldsymbol{t}^{(c)}) =
> > >     \begin{cases}
> > >      \frac{ \pi_ i \,\boldsymbol{P}_ {I}(t^c_ i; \boldsymbol{\Theta}_ {\cdot i}) }{ \pi_ i \,\boldsymbol{P}_ {I}(t^c_ i; \boldsymbol{\Theta}_ {\cdot i})
> > >      + (1-\pi_ i) \,\boldsymbol{P}_ {I}(t^c_ i; \boldsymbol{\Psi}_ {\cdot i})}, & \text{if } t_ i^c < T, \\\\
> > >      \frac{\pi_ i \,\boldsymbol{P}_ {U}(t^c_ i; \boldsymbol{\Theta}_ {\cdot i}) }{ \pi_ i \,\boldsymbol{P}_ {U}(t^c_ i; \boldsymbol{\Theta}_ {\cdot i})
> > >      + (1-\pi_ i) \,\boldsymbol{P}_ {U}(t^c_ i; \boldsymbol{\Psi}_ {\cdot i})}, & \text{if } t_ i^c \geq T.
> > >     \end{cases}
> > > \end{align*}
> > >
> > > Each $t_i^c$ is a single randomly drawn sample. Since $\hat{\pi}_i^c$ is a function of $t_i^c$, the randomness from $t_i^c$ prevents one from constructing a confidence interval for $\hat{\pi}_i^c$.
> > >
> > > Instead, we elaborate on estimating uncertainty in $\pi_i$ via a bootstrap procedure. In a small follow-up experiment, we generated a network of size $N=100$ and drew $M=2000$ samples. First, we ran our algorithm to obtain the point estimates $\\{\hat{\pi}_ i\\}_ {i=1}^N$. Next, we created 500 bootstrap replicates by resampling $M$ observations with replacement, and for each replicate $b\in\\{1,\dots,500\\}$ we recomputed the estimates $\\{\hat\pi_ i^{(b)}\\}_ {i=1}^N$. To illustrate, we order the truth $\{\pi_i\}$ in ascending order, identify the 0th, 1st, 2nd, 3rd, and 4th quartile indices of $\\{\pi_ i\\}$ (denoted as $\pi_{i\,(k)}$, $k=0,\dots,4$), and summarize the bootstrap distributions for those five $\pi_i$. Finally, for each representative $\pi_i$ we can construct an empirical $(1-\alpha)$-level confidence interval using the $(\alpha/2)$th and $(1-\alpha/2)$th percentiles of its 500 bootstrap estimates.
> > >
> > > | Representative $\boldsymbol{\pi_{i}}$                      | $\pi_{i(0)}$ | $\pi_{i(1)}$ | $\pi_{i(2)}$ | $\pi_{i(3)}$ | $\pi_{i(4)}$ |
> > > | :--------------------------------- | :----------: | :----------: | :----------: | :----------: | :----------: |
> > > | True $\pi_i$                   |     0.302    |     0.394    |     0.498    |     0.595    |     0.689    |
> > > | Estimated $\hat\pi_i$          |     0.266    |     0.332    |     0.442    |     0.589    |     0.625    |
> > > | Bootstrap 5 % percentile           |     0.005    |     0.237    |     0.403    |     0.534    |     0.571    |
> > > | Bootstrap 25 % percentile          |     0.253    |     0.259    |     0.424    |     0.560    |     0.596    |
> > > | Bootstrap 50 % percentile (median) |     0.265    |     0.286    |     0.435    |     0.578    |     0.611    |
> > > | Bootstrap 75 % percentile          |     0.277    |     0.309    |     0.450    |     0.591    |     0.626    |
> > > | Bootstrap 95 % percentile          |     0.293    |     0.344    |     0.468    |     0.608    |     0.653    |
> > > | Bootstrap standard deviation       |     0.017    |     0.080    |     0.020    |     0.023    |     0.025    |
> > >
> > > **Table 1**: Bootstrap Summaries (Rows) for Five Representative $\pi_ i$ (Columns)
> > >
> > > **Additional Notes on Q2**
> > >
> > > We’d also like to highlight that our E-step admits a closed-form solution, allowing for very efficient computation. Moreover, both the E-step and M-step are implemented using batched PyTorch tensor operations, which are vectorized across samples and executed in parallel on the GPU.

---

### Official Review · Reviewer_q7jT · 2025-07-03

**Clarity:** 2
**Significance:** 2
**Originality:** 3
**Rating:** 4
**Confidence:** 5

**Summary:**

This paper proposes a novel double network mixture model to infer hidden diffusion networks when information spread in terms of cascading behavior is highly heterogenous. Authors provided theoretical basis and conducted several experiments to support their idea.

**Questions:**

Questions:
1. As mentioned before, I am quite concerned about the scalability of the proposed method. Would it be easy to extend the method so that it could be applied to networks with at least thousands of nodes if not millions.
2. Is it possible to extend the method so that it instead learns structures rather than making hard assumptions such as low-rank latent networks?
3. Transmission rates are assumed over static networks. Can they be generalized to dynamic setting as seen in real world where nodes’ behavior may change over time?

**Ethical Concerns:**

["NO or VERY MINOR ethics concerns only"]

**Final Justification:**

I have read the rebuttal and seen their new experiments that have convinced me their work is ready for publication.

**Limitations:**

yes

**Quality:**

2

**Strengths And Weaknesses:**

Strengths:
1. The core idea of the paper, i.e., double network mixture model and distributional mixture representation of cascades is novel and it allows the model to capture complex diffusion patterns between observed and latent channels. The paper reads well and the topic is interesting.
2. Authors have provided rigorous theoretical basis to back the proposed method that guarantees the algorithm will eventually converge.

Weaknesses:
1. As also acknowledged by the authors, due to the presence of EM step, the proposed method is computationally expensive and this limits scalability of their algorithm to larger networks (N=200 is significantly smaller than most of the synthetic and real-world networks.
2. Several strong assumptions make the paper less reliable IMO. Examples include but not limited to imposing low-rank assumption on the latent network.
3. Accurate infection times are usually hard to obtain in real-world scenarios and the method seems to require precise infection times.

---

> ### Author Rebuttal · Authors · 2025-07-31
>
> We really appreciate the reviewer’s thoughtful feedback, which has significantly improved our paper! In the following response, we discuss and justify the implications of our assumptions, and demonstrate our method's generalizability with extensive experiments.
>
> **Q1**
>
> We agree with the reviewer that EM algorithm is not very scalable in general. However, although our optimization is formulated as EM-type update, we argue that the implicit convex nature of our objective function allows optimization to be as efficient as gradient descent, which is fundamentally different to classic EM, and therefore scalable to large network setting.
>
> The first reason for the poor scalability of classic EM algorithm is the posterior inference in the E-step, which may suffer from low computational efficiency due to sampling and numerical integration. However, as we illustrate in section 3.4, E-step in our algorithm has an explicit form in equation (10), and therefore can be efficiently calculated.
>
> The second scalability issue of classic EM is that the optimization in M-step may be expansive. However, we have proved in Theorem 3.1 that the objective function $\boldsymbol{Q}(\boldsymbol{\Omega})$ in the M-step is strictly concave, which makes the optimization as efficient as gradient descent. In addition, in each EM-iteration we only perform gradient ascent once instead of maximizing $\boldsymbol{Q}(\boldsymbol{\Omega})$ until convergence, where $\boldsymbol{\Omega} = \\{\boldsymbol{\Theta},\boldsymbol{\Psi},\boldsymbol{\pi}\\}$ are our model parameters.
> It is been proved in [1] that the above EM-update is equivalent to gradient decent given strictly concave and well-behaved $\boldsymbol{Q}(\boldsymbol{\Omega})$, and therefore it can be proved that our optimization enjoys the geometric convergence rate as
> \begin{align*}
>     \\|  \boldsymbol{\Omega}^{(s)} - \boldsymbol{\Omega}^{\star} \\| \leq \rho^s \\|  \boldsymbol{\Omega}^{(0)} - \boldsymbol{\Omega}^{\star}\\|,
> \end{align*}
> for some $\rho<1$, where $\boldsymbol{\Omega}^{(s)} $ are the estimation from the $s$-th update, and $\boldsymbol{\Omega}^{\star}$ are the true parameter values. The rate equivalently fast to that of gradient descent.
>
> Last but not least, when handling large networks, we can replace the standard SVD operation in the $M$-step by randomized SVD or Lanczos algorithm. Then, we can further reduce the computational cost from $\mathcal{O}(N^3)$ to $\mathcal{O}(rN^2)$ where $r<<N$ is the rank of network $\boldsymbol{\Psi}$ and $N$ is the size of network (or in practice, $r$ + $k$ where $k$ is usually 10 to 20 for more stable convergence). This is at the same order of matrix multiplication between two low-rank matrix. The overall computational complexity of our algorithm is: $T \cdot O(N^2 \cdot M) + T \cdot O(N^2 \cdot r)$, where $T$ is the number of iterations executed and $M$ is sample size.
>
> We also perform numerical experiments to justify the above theoretical perspective. To handle possible memory constraints when network size is large, we use a mini-batch version of our algorithm such that each update of both E-step and M-step are conducted using $B$ samples where $B$ is batch size. In on complete iteration, we make $\lceil M/B \rceil$ updates so that all $M$ samples are fully used.
>
> In our first experiment, we fix total number of iterations $T = 100$, sample size $M = 5000$, batch size $B = 40$, dimension $r + k = 20$ for randomized SVD, and only change $N = 500, 1000, 2000, 4000$. We report the mean and the standard deviation of execution time in second per iteration for all four different network sizes in the following table.
>
> | Size          | 500         | 1000        | 2000        | 4000        |
> |:-------------:|:-----------:|:-----------:|:-----------:|:-----------:|
> | Time (s/iter) | 0.032(0.002)| 0.952(0.004)| 3.685(0.001)| 14.574(0.006)|
>
> **Table 1** Execution Time Per Iteration under Different Node Sizes (mean ± std).
>
> In our second experiment, we fix network size $N = 1000$, sample $M = 20000$, and batch size $B = 300$. We report the execution time and performance of our algorithm against NetRate[2] when $T = 50, 75, 100$ to demonstrate the performance and scalability of our algorithm, and to give a rough estimate of how much iterations are required for ideal performance.
>
> | Algorithm | Metric                             | $T = 50$       | $T = 75$       | $T = 100$      |
> |:---------:|:----------------------------------:|:--------------:|:--------------:|:--------------:|
> | Our Alg   | Time (s)                           | 181.389(0.563) | 270.689(0.491) | 373.010(0.718) |
> |           | $MAE_{\\boldsymbol{\\Theta}}$       | 0.222(0.001)   | 0.212(0.001)   | 0.207(0.001)   |
> |           | $MAE_{\\boldsymbol{\\Psi}}$         | 0.690(0.002)   | 0.702(0.001)   | 0.693(0.001)   |
> |           | $MAE_{\\pi}$                       | 0.061(0.001)   | 0.062(0.001)   | 0.063(0.002)   |
> |           | $Acc_{\\boldsymbol{\\Psi}}$         | 0.669(0.005)   | 0.730(0.005)   | 0.773(0.003)   |
> | NetRate   | Time (s)                           | 104.849(0.010) | 156.660(0.062) | 208.546(0.109) |
> |           | $MAE_{\\boldsymbol{\\Theta}}$       | 0.870(0.002)   | 0.876(0.001)   | 0.880(0.002)   |
> |           | $MAE_{\\boldsymbol{\\Psi}}$         | 0.841(0.001)   | 0.842(0.001)   | 0.843(0.001)   |
> |           | $Acc_{\\boldsymbol{\\Psi}}$         | 0.501(0.013)   | 0.502(0.015)   | 0.493(0.018)   |
>
> **Table 2** Performance of Our Method and NetRate under Different Total Iterations (mean ± std).
>
> By the table, the ratio between our algorithm's execution time and that of NetRate [2] is a constant independent of $N$. This demonstrates the scalability of our algorithm since NetRate [2] is can be scaled efficiently to large networks. Also, our algorithm achieves better performance of that of NetRate [2].
>
> **Q2**
>
> First, we argue that low-rank assumption is not hard assumption since it is an intrinsic properties of many real-world networks. one can interpret $\boldsymbol{\Psi}$ as the diffusion over latent relations among the units that are not captured by the structural network $\boldsymbol{\Theta}$. The magnitude of $\boldsymbol{\Psi}_{ij}$ reflects the distance between unit $i$ and $j$ in terms of their latent factors. It has been found that large-scale networks typically has or can be approximated by low-rank structure [4][5][6] due to that high-dimensional latent factors can be always embeded into a low dimensional latent space that preserve original distances [4].
>
> Second, our method does not require network $\boldsymbol{\Psi}$ to be exact low-rank. For example, our method works even when all singular values are non-zero as long as the singular values decays quickly. In many real-world networks, the singular values indeed decays at exponential rate [3]. To demonstrate this point, we perform numerical experiments to show our method still recovers diffusion network violating low-rank assumption.
>
> In this experiment, we consider network size $N = 1000$, sample $M = 40000$, and batch size $B = 300$. We try to estimate a true latent network $\bf{\Psi}$ whose first 500 singular values decay at a factor of $e^{-1}$ and the others are close to 0. We report the result in the following table.
>
> | Algorithm | $MAE_{\\boldsymbol{\\Theta}}$ | $MAE_{\\boldsymbol{\\Psi}}$ | $Acc_{\\boldsymbol{\\Psi}}$ |
> |:---------:|:-----------------------------:|:---------------------------:|:---------------------------:|
> | Our Alg   | 0.259(0.001)                  | 0.446(0.002)                | 0.903(0.010)               |
> | NetRate   | 0.768(0.007)                  | 0.769(0.002)                | 0.639(0.003)               |
>
> **Table 3** Performance of Our Method and NetRate when Low Rank Assumption is Violated (mean ± std).
>
> Finally, we impose sparse structure mainly for identifying different diffusion networks. The assumption can be droped as long as we have auxiliary information to avoid ambiguity from column permutation across different  networks. For example, we can typically access $p$-dimensional features from each unit as $\\{\boldsymbol{X}_ i = (X_ {i1},\cdots, X_ {ip} )\\}_ {i=1}^N$
> . Then we can model $p$ different diffusion networks $\boldsymbol{A}^{(k)} \in \mathbb{R}_+^{N\times N}\, k = 1,\cdots,p$ as
>
> $$ (\boldsymbol{A}^{(k)})_ {ij} = f\big(\text{sim}( X_ {ik}, X_ {jk}), \alpha_k \big),\; 1\leq i \neq j \leq N,\; k = 1,\cdots, p, $$
>
> where sim$(\cdot)$ can be feature similarity measurement between unit $i$ and $j$, and $f(\cdot ; \alpha_k)$ can be any function modeling the pairwise transmission rates in the $k$-th diffusion network $\boldsymbol{A}^{(k)}$. Intuitively, $\boldsymbol{A}^{(k)}$ can be interpreted as pathways via the proximity over the $k$-th features of units. Then the multiple diffusion networks $\boldsymbol{A}^{(k)},k =1,\cdots,p$ are identifiable. In addition, when we select model $f(\cdot ; \alpha_k)$ is convex in terms of parameters $\alpha_k$, then the objective function is still convex, and therefore enjoy both the statistical and computational advantages of the original model.
>
> **Q3**
>
> The proposed method is also straightforward to extend to time-varying inference by introducing time-varying weights $w_c(t)$ that penalize old cascades, i.e.,the older a cascade $c$, the smaller its $w_c(t)$. Then the corresponding time-varying diffusion networks and mixing parameters become $\boldsymbol{\Theta}(t),\boldsymbol{\Psi}(t), \boldsymbol{\pi}(t)$, both of which smoothly changes over time and that recent cascades have higher importance in determining current network structure than old cascades.
>
> References:
>
> [1] Statistical guarantees for the EM algorithm: From population to sample-based analysis
>
> [2] Uncovering the Temporal Dynamics of Diffusion Networks
>
> [3] The low-rank hypothesis of complex systems
>
> [4] Why are big data matrices approximately low rank?
>
> [5] The link-prediction problem for social networks.
>
> [6] Link prediction via matrix factorization

---

> > ### Comment · Reviewer_q7jT · 2025-08-04
> > **Raise my score**
> >
> > I really appreciate the time authors have put to address my concerns by performing new sets of experiments. After reading authors responses to all reviewers' concerns, I'm happy to raise my score to borderline accept.

---

> > > ### Author Response · Authors · 2025-08-04
> > >
> > > Dear Reviewer q7jT,
> > >
> > > Thank you for taking the time to read our rebuttal. We greatly appreciate your constructive comments and are pleased to see the score increase upon your reconsideration. Your feedback has helped us improve the clarity and rigor of our work.

---

### Official Review · Reviewer_fDKS · 2025-07-04

**Clarity:** 3
**Significance:** 3
**Originality:** 3
**Rating:** 4
**Confidence:** 4

**Summary:**

This paper proposes a novel double network mixture model for inferring latent diffusion networks in heterogeneous cascade processes. The model represents cascade pathways as a distributional mixture of two networks, capturing complementary diffusion patterns without modeling complex individual states. The authors develop a data-driven optimization approach using EM algorithm with convex relaxations, supported by theoretical guarantees on identifiability and convergence. Experiments on synthetic and real-world data (U.S. university research topic cascades) demonstrate the model’s superiority over baselines in recovering diffusion networks and handling heterogeneity.

**Questions:**

1. For **Weakness 1**, how might the model be extended to learn or validate a **flexible, nonparametric baseline** hazard $h(\cdot)$ rather than assuming one of the fixed Exp/Pow/Ray forms?

2.  For **Weakness 2**, can the authors provide detailed scaling benchmarks for networks with **$N>200$, and discuss strategies** (e.g. distributed SVD, stochastic updates) to mitigate the polynomial—or effectively exponential—growth in the EM M-step?

3.  For **Weakness 3**, is it possible to generalize the **binary indicator $Z^c_i\in\{0,1\}$ to a time-varying or multinomial mixing variable $Z^c_i(t)$** so that nodes can switch networks mid-cascade or leverage more than two diffusion channels?

4. For **Weakness 4**, have the authors empirically verified that

$$
\frac{\partial^2}{\partial\lambda^2}H\bigl(t\mid t',\lambda\bigr)\;\ge\;0
$$

for **Pow and Ray hazards across the $\lambda$ values learned in their experiments**, to ensure the M-step convexity conditions truly hold in practice?

**Ethical Concerns:**

["NO or VERY MINOR ethics concerns only"]

**Final Justification:**

The authors provided reasonable responses to the questions I raised and supplemented them with corresponding experiments. In their response, they addressed and clarified **Theorem 3.1’s assumption of the convexity of the hazard function** and the **computational complexity of the EM algorithm**, and further noted that in the picked experiments in Section 4.1, they numerically verified that

$$
\frac{\partial^2 H(t \mid t^{\prime}, \lambda)}{\partial \lambda^2}
$$

for the learned $\lambda$ across the Pow/Ray models in these experiments is 0. This resolves the issues I mentioned in my review, and therefore, I am willing to raise my score to 4.

**Limitations:**

yes

**Quality:**

3

**Strengths And Weaknesses:**

Strengths:
This paper demonstrates **theoretical rigor** by ensuring identifiability through structural constraints (sparsity and low-rank assumptions) and offering statistical and computational guarantees via convex optimization. The column-wise network mixture effectively captures **diffusion patterns at an exponential scale**, outperforming single-network baselines in heterogeneous environments. **Extensive experiments** on synthetic datasets with varying topologies and transmission models, as well as on real research-topic cascades, convincingly validate the method’s effectiveness.

Weaknesses:
1. While the paper illustrates exponential (Exp), power‐law (Pow) and Rayleigh (Ray) models, it really only requires an additive‐risk form
 $$
       S(t\mid t_j,\lambda_{ji})= \exp\Bigl(-\int_{t_j}^{t}\lambda_{ji}h(s - t_j)\mathrm{d}s\Bigr).
     $$


for some **nonnegative baseline $h(\cdot)$**. There is no discussion of **choosing or fitting a more flexible, nonparametric $h(\cdot)$ to capture**, for example, time‐varying or abrupt changes in hazard.

2. The EM algorithm’s computational complexity grows exponentially with network size N; for N > 200, the number of mixture patterns (2^N) may **preclude convergence**, as demonstrated by experiments limited to N = 200. Moreover, enforcing a low-rank constraint on Ψ via nuclear-norm relaxation is suboptimal when the true latent network is high-rank, potentially introducing bias in dense scenarios.

3. The model fixes each node $i$ to one network per cascade ($Z^c_i\in\{0,1\}$), so it **cannot capture mid‐cascade switches or interactions among three or more diffusion channels**.

4. In the paper, Theorem 3.1 requires the hazard function to satisfy the convexity condition(line 192)

$$
\frac{\partial^2}{\partial\lambda^2}H\bigl(t\mid t',\lambda\bigr)\\ge0.
$$

However, while the Exp, Pow, and Ray models typically meet this requirement, the authors do not verify it over the actual $\lambda$ values learned in their experiments. If this convexity assumption is violated, the M-step convergence guarantees may no longer hold. **I recommend numerically evaluating**

$$
\frac{\partial^2 H(t\mid t',\lambda)}{\partial\lambda^2}
$$

for the Pow and Ray hazard functions across the trained $\lambda$ range to confirm that the condition indeed holds in practice.

---

> ### Author Rebuttal · Authors · 2025-07-31
>
> We really appreciate the reviewer’s thoughtful feedback, which has significantly improved our paper! In the following response, we discuss and justify the implications of our assumptions. We also conduct extensive experiments to demonstrate the generalizability of our method.
>
> **Q1**
>
> The transmission model in this paper can be readily generalized to nonparametric model such as kernel function following [3]. Then we will add additional layer of optimization in the M-step for updating the nonparametric model. However, it is found that Exp/Pow/Ray forms can efficiently capture the activation time lag in many real cascade [2]. In addition, the focus of our paper is to model the complex cascade mixing pattern and recover the underlying diffusion networks, instead of fitting the observed activation time lags. Based on the above reasons, we use Exp/Pow/Ray model in this paper, and leave the extension to the nonparametric transmission model as a future study.
>
> **Q2**
>
> We agree that EM algorithm is not very scalable in general. However, although our optimization is formulated as EM-type update, we argue that the implicit convex nature of our objective function allows the optimization to be as efficient as gradient descent, which is fundamentally different to classic EM, and therefore scalable to large network setting.
>
> The first reason for the poor scalability of classic EM algorithm is the posterior inference in the E-step, which may suffer from low computational efficiency due to sampling and numerical integration. However, as we pointed out in section 3.4, the E-step in our algorithm has an explicit form in equation (10), and therefore can be efficiently calculated without using parallel computing.
>
> The second scalability issue of classic EM is that the optimization in M-step may be expansive. However, we have proved in Theorem 3.1 that the objective function $\boldsymbol{Q}(\boldsymbol{\Omega})$ in the M-step is strictly concave, which makes the optimization as efficient as gradient descent. In addition, in each EM-iteration we only perform gradient ascent one time instead of maximizing $\boldsymbol{Q}(\boldsymbol{\Omega})$ until convergence, where $\boldsymbol{\Omega} = \\{\boldsymbol{\Theta},\boldsymbol{\Psi},\boldsymbol{\pi}\\}$ are our model parameters.
> It is been proved in [1] that the above EM-update is equivalent to gradient decent given strictly concave and well-behaved $\boldsymbol{Q}(\boldsymbol{\Omega})$, and therefore it can be proved that our optimization enjoys the geometric convergence rate as
>
> $$
> \\|  \boldsymbol{\Omega}^{(s)} - \boldsymbol{\Omega}^{\star} \\| \leq \rho^s \\|  \boldsymbol{\Omega}^{(0)} - \boldsymbol{\Omega}^{\star}\\|,
> $$
>
> for some $\rho<1$, where $\boldsymbol{\Omega}^{(s)} $ are the estimation from the $s$-th update, and $\boldsymbol{\Omega}^{\star}$ are the true parameter values. The rate equivalently fast to that of gradient descent.
>
> Last but not least, when handling large networks, we can replace the standard SVD operation in the $M$-step by randomized SVD or Lanczos algorithm. Then, we can further reduce the computational cost from $\mathcal{O}(N^3)$ to $\mathcal{O}(rN^2)$ where $r<<N$ is the rank of network $\boldsymbol{\Psi}$ and $N$ is the size of network (or in practice, $r$ + $k$ where $k$ is usually 10 to 20 for more stable convergence). This is at the same order of matrix multiplication between two low-rank matrix. \textbf{The overall computational complexity of our algorithm is: $T \cdot O(N^2 \cdot M) + T \cdot O(N^2 \cdot r)$}, where $T$ is the number of iterations executed and $M$ is sample size.
>
> Besides the above justification of our method's scalability from a theoretic perspective, we also perform numerical experiments to proof it. To handle possible memory constraints when network size is large, we use a mini-batch version of our algorithm such that each update of both E-step and M-step are conducted using $B$ samples where $B$ is batch size. In on complete iteration, we make $\lceil M/B \rceil$ updates so that all $M$ samples are fully used.
>
> In our first experiment, we fix total number of iterations $T = 100$, sample size $M = 5000$, batch size $B = 40$, dimension $r + k = 20$ for randomized SVD, and only change $N = 500, 1000, 2000, 4000$. We report the mean and the standard deviation of execution time in second per iteration for all four different network sizes in the following table.
>
> | Size          | 500         | 1000        | 2000        | 4000        |
> |:-------------:|:-----------:|:-----------:|:-----------:|:-----------:|
> | Time (s/iter) | 0.032(0.002)| 0.952(0.004)| 3.685(0.001)| 14.574(0.006)|
>
> **Table 1** Execution Time Per Iteration under Different Node Sizes (mean ± std).
>
> We see that when network size $N$ is large enough for GPU kernels to start fully synchronize, the execution time starts to agree with the theoretical computation complexity of our algorithm: As $N$ doubles, execution time per iteration increases by a factor of four. Note that if more memory is available, we can increase the batch size $B$ accordingly for better parallelization on GPU and thus shorter execution time per iteration. Please see response to reviewer 3 for more numerical experiments related to the scalability of our method.
>
> **Q3**
>
> In this paper, we impose sparse and low-rank structure to differentiate diffusion networks. In general, the proposed method can be extended to multiple diffusion networks as long as we have auxiliary information to avoid ambiguity from column permutation across different  networks. For example, we can typically access $p$-dimensional features from each unit as $\\{\boldsymbol{X}_ i = (X_ {i1},\cdots, X_ {ip} )\\}_ {i=1}^N$. Then we can model $p$ different diffusion networks $\boldsymbol{A}^{(k)} \in \mathbb{R}_+^{N\times N}\, k = 1,\cdots,p$ as
>
> $$ (\boldsymbol{A}^{(k)})_ {ij} = f\big(\text{sim}( X_ {ik}, X_ {jk}), \alpha_k \big),\; 1\leq i \neq j \leq N,\; k = 1,\cdots, p, $$
>
> where sim$(\cdot)$ can be feature similarity measurement between unit $i$ and $j$, and $f(\cdot ; \alpha_k)$ can be any function modeling the pairwise transmission rates in the $k$-th diffusion network $\boldsymbol{A}^{(k)}$. Intuitively, $\boldsymbol{A}^{(k)}$ can be interpreted as pathways via the proximity over the $k$-th features of units. Then the multiple diffusion networks $\boldsymbol{A}^{(k)},k =1,\cdots,p$ are identifiable. In addition, when we select model $f(\cdot ; \alpha_k)$ is convex in terms of parameters $\alpha_k$, then the objective function is still convex, and therefore enjoy both the statistical and computational advantages of the original model.
>
> The proposed method is also straightforward to extend to time-varying inference by introducing time-varying weights $w_c(t)$ that penalize old cascades, i.e.,the older a cascade $c$, the smaller its $w_c(t)$. Then the corresponding time-varying diffusion networks and mixing parameters become $\boldsymbol{\Theta}(t),\boldsymbol{\Psi}(t), \boldsymbol{\pi}(t)$, both of which smoothly changes over time and that
> recent cascades have higher importance in determining current network structure than old cascades.
>
>
> **Q4**
>
> Since we only use Exp/Pow/Ray model as explained in our response to the first question, $\partial^2H(t |t^{\prime}, \lambda)/\partial\lambda^2$ is 0 and does not depend on $t$ or $\lambda$ for all three models. We have also picked experiments in Section 4.1 and numerically verified that $\partial^2H(t |t^{\prime}, \lambda)/\partial\lambda^2$ for learned $\lambda$ across Pow/Ray models in these experiments are 0.
>
> References:
>
> [1] Statistical guarantees for the EM algorithm: From population to sample-based analysis
>
> [2] Uncovering the structure and temporal dynamics of information propagation
>
> [3] Learning networks of heterogeneous influence

---

> > ### Comment · Reviewer_fDKS · 2025-08-05
> > **Thank you for your detailed response！**
> >
> > Thank you for your detailed responses to the concerns I raised. Thanks to your explanations, I have gained a deeper understanding of both the assumption of the hazard function’s convexity in Theorem 3.1 and the computational complexity of the EM algorithm. I hope you will incorporate these improvements into the paper, as they will greatly enhance its validity. Consequently, I have increased my scores.

---

> > > ### Author Response · Authors · 2025-08-05
> > >
> > > Dear Reviewer fDKS,
> > >
> > > Thank you for reviewing our rebuttal. We appreciate your constructive feedback and are delighted that our revisions led to a higher score. Your comments have helped us sharpen the clarity and rigor of our work, and we will incorporate these suggestions into the final manuscript.
> > >
> > > If our earlier mention of “Reviewer 3” caused any confusion—since we were referring to Reviewer q7jT—and if you are still interested in the additional scalability experiments, please see our responses to Q1 and Q2 for Reviewer q7jT. We apologize for any inconvenience and invite you to review those results at your convenience.

---

### Official Review · Reviewer_3jm1 · 2025-07-16

**Clarity:** 3
**Significance:** 4
**Originality:** 3
**Rating:** 4
**Confidence:** 4

**Summary:**

The paper proposes a method for inferring heterogeneous diffusion structure from network cascades. In particular, cascades can follow different diffusion pathways in a multi-layer network topology, where each transmission channel has a different transmission rate. Motivated by this, they propose to infer transmission rates and topologies for both the observed network (for instance a social network) and the latent diffusion network (based on latent social factors), and individual node probabilities that cascades follow these channels. The parameters of the resulting double network mixture model are constrained using sparsity and low-rank constraints, and optimized using the EM algorithm. Guarantees for convexity of the parameter sets are also obtained. Results are reported on three synthetic datasets where the proposed method outperforms existing work, as well as on a dataset of US social science programs where the method recovers meaningful topic diffusion networks.

**Questions:**

1. Please compare to references [A,B]?
2. Discuss the applicability of the method to the case where more than two network pathways are involved?
3. Provide timing analysis to address concerns about scalability?
4. These are more for clarification- in Proposition 2, line 165, page 5, how does a larger value of the matrix 2-norm indicate lower rank?
Likewise, in line 166, how does a larger value of the maxnorm indicate a lower sparsity level? Definition of accuracy in line 210 on page 7, how is higher the better?

**Ethical Concerns:**

["NO or VERY MINOR ethics concerns only"]

**Final Justification:**

The concerns I raised in the review were mostly addressed by the authors' rebuttal. The scalability issue remains, as also pointed out by other reviewers, at the same time, the low rank constraint which causes this scalability problem is arguably necessary for model identification. Given the methodological contributions and rigor of presentation outweigh the scalability downside by some margin, I would recommend a borderline accept rating.

**Limitations:**

Scalability of the method is a concern.

**Paper Formatting Concerns:**

N.A.

**Quality:**

4

**Strengths And Weaknesses:**

Strengths:
1. The paper is well written and easy to follow along.
2. The problem of inferring heterogeneous diffusion network structure from cascade data is interesting.
3. The proposed method's usefulness is demonstrated empirically.

Weaknesses:
1. Missing citations to more recent work on inferring network structure from cascades, [A, B].
2. The case where more than two transmission channels may be involved should be discussed as well for a comprehensive treatment.
3. As admitted by the authors, the method is not scalable beyond networks of a few hundred nodes. This limits its usefulness in a real epidemiological scenario where interventions are required.

[A] Rong, Yu, Qiankun Zhu, and Hong Cheng. "A model-free approach to infer the diffusion network from event cascade." Proceedings of the 25th ACM international on conference on information and knowledge management. 2016.
[B] Gray, Caitlin, Lewis Mitchell, and Matthew Roughan. "Bayesian inference of network structure from information cascades." IEEE Transactions on Signal and Information Processing over Networks (2020).

---

> ### Author Rebuttal · Authors · 2025-07-31
>
> We really appreciate the reviewer’s thoughtful feedback, which has significantly improved our paper! In the following response, we discuss and justify the implications of our assumptions, and demonstrate our method's generalizability with extensive experiments.
>
> **Q1**
>
> We have thoroughly examined both the Bayesian‐MCMC framework [B] and the NPDC algorithm [A]. Although the Bayesian MCMC framework [B] can jointly infer both network topology and transmission‐rate posteriors under general diffusion models including Exp/Pow/Ray models, its computational cost is high. NPDC is a completely model‐free method that only infers network topology from a clustering perspective. Both methods thus support very general diffusion dynamics, yet neither was designed to handle the specific scenario we target—a mixture of propagation pathways spanning two overlapping networks. Due to their similarity in this way and time constraint, we limit our empirical evaluation to an illustrative comparison against NPDC [A].
>
> In this experiment, we consider network size $N = 100$ and sample $M = 2000$. We try to estimate a true latent network $\bf{\Psi}$ whose first 75 singular values decay at a factor of $e^{-1}$ and the others are close to 0. We report the result in the following table.
>
> | Algorithm | $MAE_{\\boldsymbol{\\Theta}}$ | $MAE_{\\boldsymbol{\\Psi}}$ | $Acc_{\\boldsymbol{\\Psi}}$ | $Pre_{\\boldsymbol{\\Psi}}$ | $Recall_{\\boldsymbol{\\Psi}}$ |
> |:---------:|:---------------------------:|:---------------------------:|:---------------------------:|:---------------------------:|:-----------------------------:|
> | Our Alg   | 0.155(0.020)               | 0.283(0.041)               | 0.862(0.042)               | 0.761(0.063)               | 1.000(0.001)                 |
> | NPDC      | –                          | –                          | 0.076(0.018)               | 0.040(0.009)               | 0.715(0.248)                 |
>
> **Table 1** Performance of Our Method and NPDC (mean ± std)
>
> Results in the table show that our method outperforms NPDC under this setting. Note that NPDC returns a score for each pair of nodes in the network that indicates how likely there exists an edge between this pair of nodes. To compare the results of NPDC under the specified metrics, these scores are converted to binary values.
>
> **Q2**
>
> In this paper, we impose sparse and low-rank structure to differentiate diffusion networks. In general, the proposed method can be extended to multiple diffusion networks as long as we have auxiliary information to avoid ambiguity from column permutation across different  networks. For example, we can typically access $p$-dimensional features from each unit as $\{\boldsymbol{X} _i = (X _{i1},\cdots, X _{ip} )\} _{i=1}^N$. Then we can model $p$ different diffusion networks  $\boldsymbol{A}^{(k)} \in \mathbb{R} _+^{N\times N},\; k = 1,\cdots,p$ as
>
> $
>  (\boldsymbol{A}^{(k)})_{ij} =  f \big( \text{sim}( X _{ik}, X _{jk}), \alpha_k \big),  \; 1\leq i \neq j \leq N,\; k = 1,\cdots, p,
> $
>
> where $\text{sim}(\cdot)$ can be feature similarity measurement between unit $i$ and $j$, and $f(\cdot ; \alpha_k)$ can be any function modeling the pairwise transmission rates in the $k$-th diffusion network $\boldsymbol{A}^{(k)}$. Intuitively, $\boldsymbol{A}^{(k)}$ can be interpreted as pathways via the proximity over the $k$-th features of units. Then the multiple diffusion networks $\boldsymbol{A}^{(k)},k =1,\cdots,p$ are identifiable. In addition, when we select model $f(\cdot ; \alpha_k)$ is convex in terms of parameters $\alpha_k$, then the objective function is still convex, and therefore enjoy both the statistical and computational advantages of the original model.
>
> **Q3**
>
> We agree that EM algorithm is not very scalable in general. However, although our optimization is formulated as EM-type update, we argue that the implicit convex nature of our objective function allows the optimization to be as efficient as gradient descent, which is fundamentally different to classic EM, and therefore scalable to large network setting.
>
> The first reason for the poor scalability of classic EM algorithm is the posterior inference in the E-step, which may suffer from low computational efficiency due to sampling and numerical integration. However, as we pointed out in section 3.4, the E-step in our algorithm has an explicit form in equation (10), and therefore can be efficiently calculated without using parallel computing.
>
> The second scalability issue of classic EM is that the optimization in M-step may be expansive. However, we have proved in Theorem 3.1 that the objective function $\boldsymbol{Q}(\boldsymbol{\Omega})$ in the M-step is strictly concave, which makes the optimization as efficient as gradient descent. In addition, in each EM-iteration we only perform gradient ascent one time instead of maximizing $\boldsymbol{Q}(\boldsymbol{\Omega})$ until convergence, where $\boldsymbol{\Omega} = \\{\boldsymbol{\Theta},\boldsymbol{\Psi},\boldsymbol{\pi}\\}$ are our model parameters.
> It is been proved in [1] that the above EM-update is equivalent to gradient decent given strictly concave and well-behaved $\boldsymbol{Q}(\boldsymbol{\Omega})$, and therefore it can be proved that our optimization enjoys the geometric convergence rate as
>
> $$
> \\|  \boldsymbol{\Omega}^{(s)} - \boldsymbol{\Omega}^{\star} \\| \leq \rho^s \\|  \boldsymbol{\Omega}^{(0)} - \boldsymbol{\Omega}^{\star}\\|,
> $$
>
> for some $\rho<1$, where $\boldsymbol{\Omega}^{(s)} $ are the estimation from the $s$-th update, and $\boldsymbol{\Omega}^{\star}$ are the true parameter values. The rate equivalently fast to that of gradient descent.
>
> Last but not least, when handling large networks, we can replace the standard SVD operation in the $M$-step by randomized SVD or Lanczos algorithm. Then, we can further reduce the computational cost from $\mathcal{O}(N^3)$ to $\mathcal{O}(rN^2)$ where $r<<N$ is the rank of network $\boldsymbol{\Psi}$ and $N$ is the size of network (or in practice, $r$ + $k$ where $k$ is usually 10 to 20 for more stable convergence). This is at the same order of matrix multiplication between two low-rank matrix. \textbf{The overall computational complexity of our algorithm is: $T \cdot O(N^2 \cdot M) + T \cdot O(N^2 \cdot r)$}, where $T$ is the number of iterations executed and $M$ is sample size.
>
> Besides the above justification of our method's scalability from a theoretic perspective, we also perform numerical experiments to proof it. To handle possible memory constraints when network size is large, we use a mini-batch version of our algorithm such that each update of both E-step and M-step are conducted using $B$ samples where $B$ is batch size. In on complete iteration, we make $\lceil M/B \rceil$ updates so that all $M$ samples are fully used.
>
> In our first experiment, we fix total number of iterations $T = 100$, sample size $M = 5000$, batch size $B = 40$, dimension $r + k = 20$ for randomized SVD, and only change $N = 500, 1000, 2000, 4000$. We report the mean and the standard deviation of execution time in second per iteration for all four different network sizes in the following table.
>
> | Size          | 500         | 1000        | 2000        | 4000        |
> |:-------------:|:-----------:|:-----------:|:-----------:|:-----------:|
> | Time (s/iter) | 0.032(0.002)| 0.952(0.004)| 3.685(0.001)| 14.574(0.006)|
>
> **Table 2** Execution Time Per Iteration under Different Node Sizes (mean ± std).
>
> We see that when network size $N$ is large enough for GPU kernels to start fully synchronize, the execution time starts to agree with the theoretical computation complexity of our algorithm: As $N$ doubles, execution time per iteration increases by a factor of four. Note that if more memory is available, we can increase the batch size $B$ accordingly for better parallelization on GPU and thus shorter execution time per iteration. Please see response to reviewer 3 for more numerical experiments related to the scalability of our method.
>
> **Q4**
>
> In Proposition 2 where a lower value for
>
> $\max _{N \in \boldsymbol{\Lambda _1}(\boldsymbol{\Theta}),\|N\| _{\infty} \leq 1}\|N\| _2$
>
> indicates a higher rank of $N$. Intuitively, for $N \in \{ \boldsymbol{\Lambda_1}(\boldsymbol{\Theta}),\|N\|_{\infty} \leq 1\}$, the Frobenius norm $\| N \|_F$ is up bounded due to support of $N$ is fixed and magnitude of each element in $N$ is also up bounded. Given the relation that
>
> $\| N \|^2_F = \sum_{i=1}^N \sigma^2_i$,
>
> if largest $\sigma_i$, $\| N \|_2$ becomes larger, then the number of zero singular values can be larger, i.e., lower rank.
> If $\| N \|_2$ become smaller, then the number of zero singular values increases, i.e., higher rank. Similarly, for term
>
> $\max_{N \in \boldsymbol{\Lambda _2}(\boldsymbol{\Psi}),\|N\| _2 \leq 1}\|N\| _{\infty}$,  consider $N \in \{\boldsymbol{\Lambda_2}(\boldsymbol{\Psi}),\|N\|_2 \leq 1\}$  basically up bounds the $\| N \|^2_F$ since
>
> $\| N \| _F \leq \sqrt{\text{rank}}\| N \| _2$.
>
> Then if $\|N\|_{\infty}$ is larger then $N$ has to be sparser.
>
> References:
>
> [1] Statistical guarantees for the EM algorithm: From population to sample-based analysis

---

> > ### Author Response · Authors · 2025-08-05
> >
> > Dear Reviewer 3jm1,
> >
> > We would like to offer an additional clarification and apologize for any potential confusion: our reference to “Reviewer 3” should have been “Reviewer q7jT.” If you are interested, the additional scalability experiments can be found in Q1 and Q2 of Reviewer q7jT’s response. Thank you for your understanding and we apologize again for any inconvenience.

---

> > > ### Comment · Reviewer_3jm1 · 2025-08-05
> > > **Increasing score**
> > >
> > > Thank you for the detailed response to my questions, it does address most of my concerns. It looks like the bottleneck in terms of scalability stems from the SVD decomposition, which in turn is due to the rank constraint. Perhaps using a different constraint or other auxiliary information can address this. I recommend that the authors include the foregoing discussion on scalability for comprehensiveness. Anyhow, I can increase my score in response to the author's efforts.

---

> > > > ### Author Response · Authors · 2025-08-07
> > > >
> > > > Dear Reviewer 3jm1,
> > > >
> > > > Thank you for taking the time to review our rebuttal and for your encouraging comments. We’re delighted that our revisions have addressed your concerns and led to a higher score.
> > > >
> > > > We fully agree that the full SVD is the main scalability bottleneck. To mitigate this, we now employ randomized SVD (or Lanczos methods), reducing its computational complexity to the same order as our gradient-descent updates. As a result, SVD no longer dominates the runtime. That said, any low-rank constraint (to facilitate identifiability) still entails extra computation via SVD. As you suggest, alternative constraints or auxiliary information that can ensure identifiability—such as known support of $\boldsymbol{\Psi}$— can replace the approximate low-rank requirement. We will include this discussion in the final manuscript for completeness.

---

### Note · Authors · 2025-08-11

Dear Area Chair and Reviewers,

Thank you for your thoughtful engagement throughout the discussion period. As it has concluded, we would like to offer a brief summary as our Author Final Remarks. Based on the exchanges, we believe we have addressed all reviewers' concerns without any unsolved questions, with scalability being the central theme. Our responses emphasize scalability both theoretically and empirically.

Theoretically, we show that the E-step admits a closed-form solution with efficient computation; the M-step has complexity comparable to a single gradient-descent update; and we employ randomized SVD/Lanczos to reduce the SVD bottleneck to gradient-step scale.

Empirically, we conduct timing studies on networks of 500, 1,000, 2,000, and 4,000 nodes, confirming runtime improvements consistent with our theoretical analysis. We also benchmark on a 1,000-node network against NetRate and find that our method outperforms it on evaluation metrics while retaining comparable runtime, even for large networks.

For the final version, we will incorporate all discussion points into the Discussion Section, placing particular emphasis on the theoretical analysis of scalability. We will also include all other points raised during the discussion period such as potential extensions to $n >2$ networks and robustness to possible violations of the low-rank assumption. If space is limited, we will move some details to the Appendix. In addition, we will include a more detailed and systematic numerical  study on scalability—building on the experiments shared during the discussion—as a new Appendix section. Finally, we will revise the writing to further improve clarity.

We hope this summary is helpful to everyone involved in the review process. Thank you all for your time, thoughtful feedback, and continued consideration.

---

### Decision · Program_Chairs · 2025-09-17

**Decision:**

Accept (poster)

**Comment:**

The paper proposes a double mixture graphical model for inferring latent diffusion networks from heterogeneous cascade data, with theoretical guarantees and empirical validation on synthetic and real-world datasets.

The work makes a contribution by introducing a novel modeling framework that captures heterogeneous diffusion patterns through overlapping network structures. The approach is well-motivated, mathematically rigorous, and supported by clear identifiability and optimization guarantees. The empirical studies, including both synthetic cascades and sociological topic diffusion data, demonstrate ability to recover meaningful latent structures and outperform baselines.

For the camera-ready version, the authors should expand the scalability analysis with clearer discussion of limitations and potential improvements, including practical constraints when networks grow larger. They should emphasize robustness to violations of the low-rank assumption and discuss how imprecise infection times may affect applicability. Adding more qualitative validation in the real-world case study and improving clarity of notation and figure captions would further strengthen the final version.